# Shared genetic and neuroimmune architecture links type 1 diabetes with neurocognitive traits

Priscilla Saarah[1,2,10], Zehra A. Syeda[1,2,10], Ziang Xu [1,2], Yikai Dong[1,2], Habei Jiang [1,2], Michelle Shanguhyia[1,2], Sourav Roy[1,2], Biqing Zhu [3], Le Zhang [4,5], Andrew T. Dewan [6,7], Samira Asgari [8,9] & David A. Alagpulinsa [1,2] ✉

Type 1 diabetes, particularly with childhood onset, is associated with altered neurocognitive traits, yet the underlying biological mechanisms are unclear. Here, we integrate genome-wide association results with single-cell epigenomic profiles and show that type 1 diabetes heritability is enriched in accessible chromatin of human brain-resident cells, most notably microglia, across neurodevelopment into adulthood. Bonferroni-corrected cross-trait genetic correlation analyses reveal negative correlations of type 1 diabetes with intelligence, executive function, and bipolar disorder, and a positive correlation with myasthenia gravis. Conjunctional false discovery rate analysis identifies pleiotropic loci jointly influencing type 1 diabetes and neurocognitive traits, including the 17q21.31 neurogenomic hub. Mendelian randomization further demonstrates protective effects of educational attainment, intelligence, Alzheimer's disease, and bipolar disorder on type 1 diabetes risk, whereas liability to multiple sclerosis and myasthenia gravis increases type 1 diabetes risk. In the reverse direction, liability to type 1 diabetes is associated with increased risk of myasthenia gravis. We identify several gene expression regulatory variants in brain and immune cells that jointly influence type 1 diabetes and neurocognitive traits, some of which show concordant differential expression in disease-affected versus control tissue. Together, these findings highlight pleiotropic genetic and neuroimmune mechanisms that link type 1 diabetes with cognition and neuropsychiatric disease risk.

Type 1 diabetes (T1D) is characterized by T cell-mediated destruction of insulin-producing pancreatic beta cells, necessitating lifelong dependence on insulin therapy to control glycemia. Despite advances in management, people with T1D lose more than a decade of life expectancy[1,2] and over two decades of healthy life[3] compared with the general population.

Cognitive deficits—which are antecedents and characteristic symptoms of neuropsychiatric disorders—are consistently more prevalent in people with T1D than in the general population[4]. Childhood-onset T1D is particularly associated with cognitive deficits, including reduced working memory, executive function (EXF), and performance IQ[5–8]. Children with T1D often underachieve educational goals compared to their peers[9–11], with even poorer outcomes in those with co-occurring psychiatric disorders[9]. Neuroimaging studies support these findings, showing altered neuroanatomical features such as reduced gray and white matter volumes in children with T1D[12–14].

Adults with T1D also exhibit accelerated cognitive decline[15,16] and smaller total brain volumes[16]. Epidemiological studies further demonstrate higher rates of psychiatric disorders in people with T1D compared to the general population[11,17–19].

Because hyperglycemia is the defining pathology of T1D, research has emphasized its impact on brain development and function, identifying correlations between poor glycemic control and cognitive or psychiatric comorbidities[8,20–23]. This view has reinforced the idea that neurocognitive deficits in T1D arise primarily as complications of dysglycemia[8,14,24–27]. T1D[28] and cognitive or neuropsychiatric disorders[29] are substantially heritable, with estimates reaching ~50% and ~80%, respectively. Despite this, whether shared genetic mechanisms contribute to their co-occurrence has not been systematically investigated.

Observational studies are prone to confounding by socio-demographic factors and reverse causality. For example, impaired EXF may reduce adherence to glycemic control and exacerbate disease outcomes. Moreover, because T1D is frequently diagnosed during childhood or adolescence, which is a critical window for neurodevelopment[24–27,30], clarifying whether shared genetic mechanisms jointly influence T1D and neurodevelopmental outcomes is essential.

In this study, we systematically investigate the shared genetic and cellular architecture between T1D and neurocognitive traits. By integrating genome-wide association data with single-cell epigenomic annotations, we demonstrate that T1D heritability is significantly enriched in accessible chromatin regions of brain-resident cells, particularly microglia. We further identify pleiotropic loci, genes, and genetically regulated expression patterns that converge on neuroimmune pathways, pointing to a shared genetic basis linking T1D with neurocognitive outcomes. Notably, these include the 17q21.31 locus, previously implicated in neurodevelopment and psychiatric disorders, which emerges as a key genomic hub of T1D-neurocognitive convergence.

## Results

### The heritability of T1D is enriched in brain-resident cells across development

Genetic variants associated with complex traits and diseases often localize to regulatory regions where they act in a cell-type-specific manner. To test whether T1D risk variants are enriched in regulatory landscapes of brain-resident cells, we applied stratified linkage disequilibrium score regression (S-LDSC)[31] to integrate fine-mapped T1D genome-wide association studies (GWAS) signals (18,942 T1D cases, 520,580 control individuals)[32] with single-nucleus Assay for Transposase-Accessible Chromatin using sequencing (snATAC-seq) profiles of human cortex across developmental stages from prenatal stage to adulthood[33]. Enrichment was defined as the proportion of T1D single-nucleus polymorphism (SNP)-heritability explained by a given chromatin annotation divided by the proportion of SNPs in that annotation; values above 1 indicate more heritability than expected by chance[31,34], with significance assessed using one-sided S-LDSC $P$-values. This analysis revealed temporally dynamic and cell-type-specific enrichment of T1D heritability, with the strongest and most consistent signal enrichment in microglia (prenatal enrichment = 5.04, $z = 1.69$, one-sided $P = 0.045$; adult enrichment = 10.80, $z = 2.11$, $P = 0.017$; Fig. 1a). Astrocytes also showed nominal enrichment in the prenatal stage, whereas other glial and neuronal lineages showed little signal.

To assess specificity, we performed the same S-LDSC analysis using GWAS for height, a highly heritable polygenic trait without known immune or neural involvement[35]. There was nominal enrichment in only vascular cells during early prenatal ($z = 1.74$, $P = 0.041$) and late postnatal ($z = 1.81$, $P = 0.035$) stages and no enrichment in any

other brain-resident cell type across all developmental stages (Supplementary Fig. 1a–e).

Single-nucleus ATAC-seq annotations used for S-LDSC aggregate peaks across many nuclei within each cell-type cluster and may under-detect transient, rare, or highly context-specific chromatin accessibility states present in only a small subset of cells. To complement these cluster-level analyzes, we next applied SCAVENGE (Single-Cell Analysis of Variant Enrichment through Network Propagation of Genomic Annotations), a network-propagation framework that infers trait-relevance scores (TRS) at single-cell resolution[36]. SCAVENGE confirmed persistent and robust enrichment of T1D heritability in microglia across all developmental stages (median TRS > 3.0–6.5) and additionally revealed highly significant, context-specific enrichment in inhibitory neurons (median TRS > 3) during the early postnatal and adult periods (Fig. 1b–e). Smaller but notable stage-specific enrichments were also observed in excitatory neurons (early postnatal) and oligodendrocyte lineage cells (adulthood). These enrichments do not imply that brain-resident cells replace the established pathogenic role of peripheral immune cells in T1D[32]; instead, they suggest that T1D-associated variants also act through regulatory programs within the brain, consistent with pleiotropic genetic architecture.

To provide proper context, we performed parallel SCAVENGE analyzes for Alzheimer's disease (AD) and bipolar disorder (BPD) GWAS variants, two traits with known brain-cell architectures[37], as positive controls, and height as a negative control. For AD, enrichment was strongest and most specific in microglia as expected, with secondary signals in vascular cells and astrocytes (Fig. 1f). For BPD, enrichment localized primarily to neurons in adulthood and to microglia at earlier developmental stages (Fig. 1g), reflecting established disease biology. In contrast, height showed strong enrichment in vascular cells (TRS > 3.0) but no significant signal in microglia, neurons, or other brain cell types (Supplementary Fig. 1b).

Overall, these complementary analyzes indicate that, in addition to the well-established enrichment in peripheral immune cells[32], T1D genetic liability also impacts regulatory programs in brain-resident cell types, with the strongest and most consistent signal enrichment observed in microglia, alongside developmental stage–specific signals in other neural and glial populations that may be incompletely captured by current single-cell genomic annotations.

### The genetic architecture of T1D overlaps with neurocognitive traits

Given the enrichment of T1D heritability in brain-resident cells, we used cross-trait LDSC to assess genome-wide genetic correlation between T1D and each of 21 phenotypes with measurable neurocognitive components (Supplementary Table 1). These traits span four neurocognitive-related domains, including (i) core cognitive and learning traits (e.g., intelligence, EXF, educational attainment), (ii) psychiatric disorders (e.g., BPD and schizophrenia), (iii) neurological conditions (e.g., Alzheimer's disease, Parkinson's disease, migraine, sleep phenotypes), and (iv) neuroautoimmune disorders (e.g., multiple sclerosis, myasthenia gravis). For all T1D-neurocognitive trait pairs, the bivariate LDSC intercepts were close to 1.0, indicating minimal residual covariance and that sample overlap is unlikely to meaningfully bias the genetic correlation estimates.

T1D showed Bonferroni-corrected significant negative genetic correlations with intelligence ($r_g = -0.09$, SE = 0.02, $P = 7.8 \times 10^{-5}$), EXF ($r_g = -0.08$, SE = 0.03, $P = 0.0013$), and BPD ($r_g = -0.11$, SE = 0.03, $P = 6.4 \times 10^{-5}$), and a positive correlation with myasthenia gravis ($r_g = 0.37$, SE = 0.08, $P = 3 \times 10^{-6}$). Nominally significant ($P < 0.05$ but not meeting Bonferroni threshold $P < 0.00238$) included negative correlations with educational attainment and autism spectrum disorder and positive correlations with migraine, insomnia, and multiple sclerosis (Fig. 2a, and Supplementary Data 1).

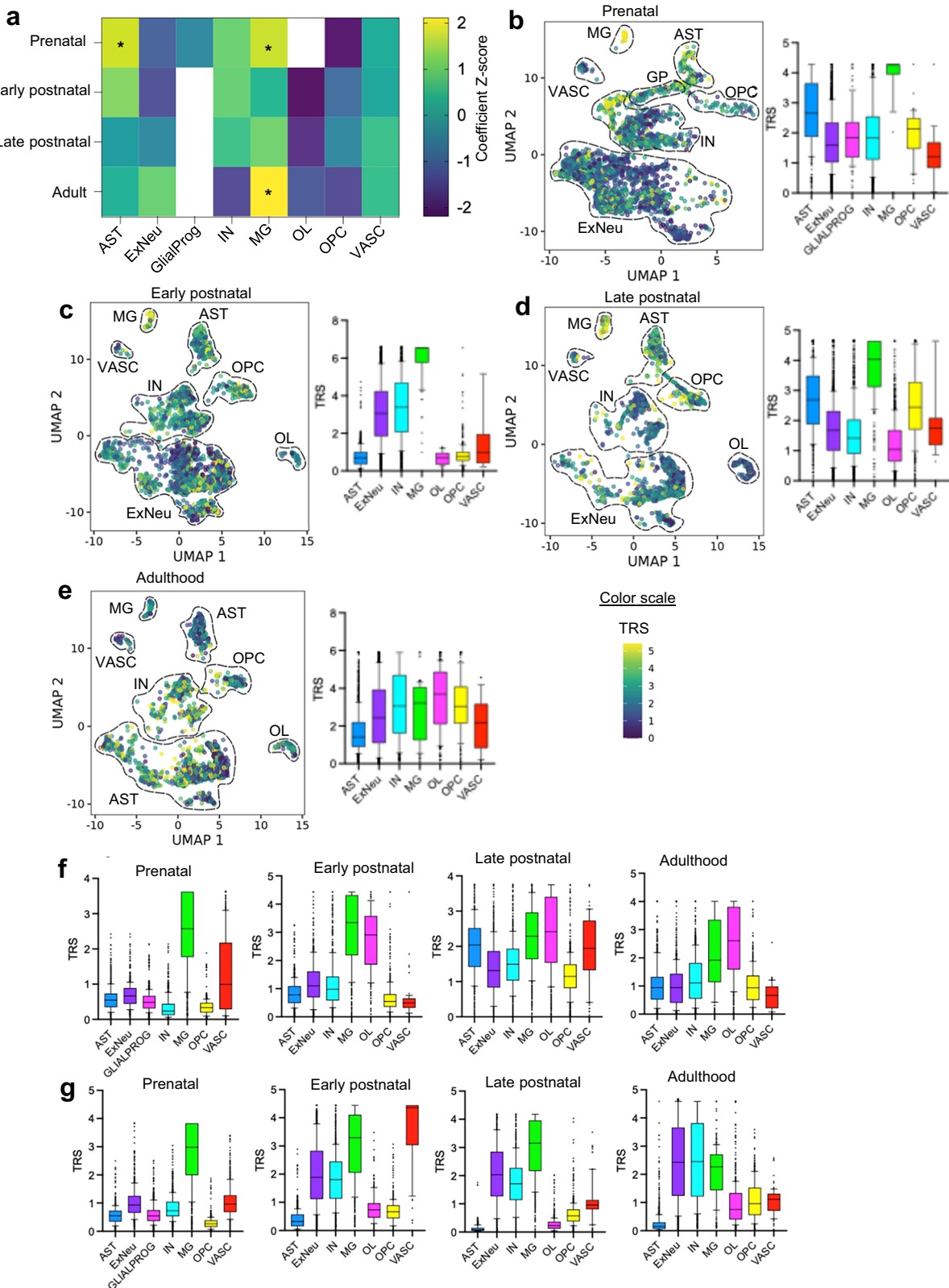

To further dissect the genetic overlap between T1D and neuro-cognitive traits, we applied MiXeR, which estimates polygenicity and quantifies the number of causal variants shared between traits. Unlike LDSC, which summarizes genome-wide correlation as an average effect direction, MiXeR captures pleiotropy irrespective of direction. MiXeR revealed extensive variant sharing between T1D and neuro-cognitive traits (Fig. 2b). MiXeR showed that T1D is moderately

polygenic (~22,000 variants) and shares hundreds of causal variants with most of the examined neurocognitive traits. The strongest overlap was with myasthenia gravis, consistent with their shared auto-immune basis, but notable overlap was also observed with intelligence and educational attainment. MiXeR analyzes revealed extensive sharing of causal variants between T1D and neurocognitive traits irrespective of effect direction. In several cases, substantial polygenic

**Fig. 1 | Enrichment of type 1 diabetes heritability in brain-resident cell types across neurodevelopment. a** Stratified LD score regression (S-LDSC) analysis of type 1 diabetes genome-wide association study variants in accessible chromatin of brain cell types–astrocytes (AST), excitatory neurons (ExNeu), glial progenitors (GLIALPROG), inhibitory neurons (IN), microglia (MG), oligodendrocyte progenitor cells (OPC), and vascular cells (VASC)–across four neurodevelopmental stages (prenatal, early postnatal, late postnatal, and adult), profiled by single-nucleus assay for transposase-accessible chromatin sequencing. The heatmap displays S-LDSC τ Z-scores (τ/SE), reflecting cell-type–specific heritability enrichment. Asterisks indicate nominal significance based on one-sided *P* values (*P* < 0.05) from stratified LD score regression. Source data are provided in Supplementary Data 9. SCAVENGE trait-relevance scores (TRS) for type 1 diabetes fine-mapped variants in single-nucleus assay for transposase-accessible chromatin sequencing data across prenatal (**b**), early postnatal (**c**), late postnatal (**d**), and adult (**e**) stages. **f** SCAVENGE TRS based on Alzheimer's disease genome-wide association study–significant variants. **g** SCAVENGE TRS based on bipolar disorder genome-wide association study–significant variants. Box plots show the median (centre line), interquartile range (IQR; box), and whiskers extending to 1.5× IQR. For SCAVENGE analyses, *n* denotes i*n*dividual nuclei from the source single-nucleus datasets. Source data are provided in Supplementary Data 11–13. Statistical analyses were performed using publicly available genome-wide association study summary statistics and single-cell reference datasets. The unit of analysis corresponds to individual study participants in the original genome-wide association studies and individual nuclei in the single-cell datasets as defined in the source studies. No biological or technical replicates were generated in this study.

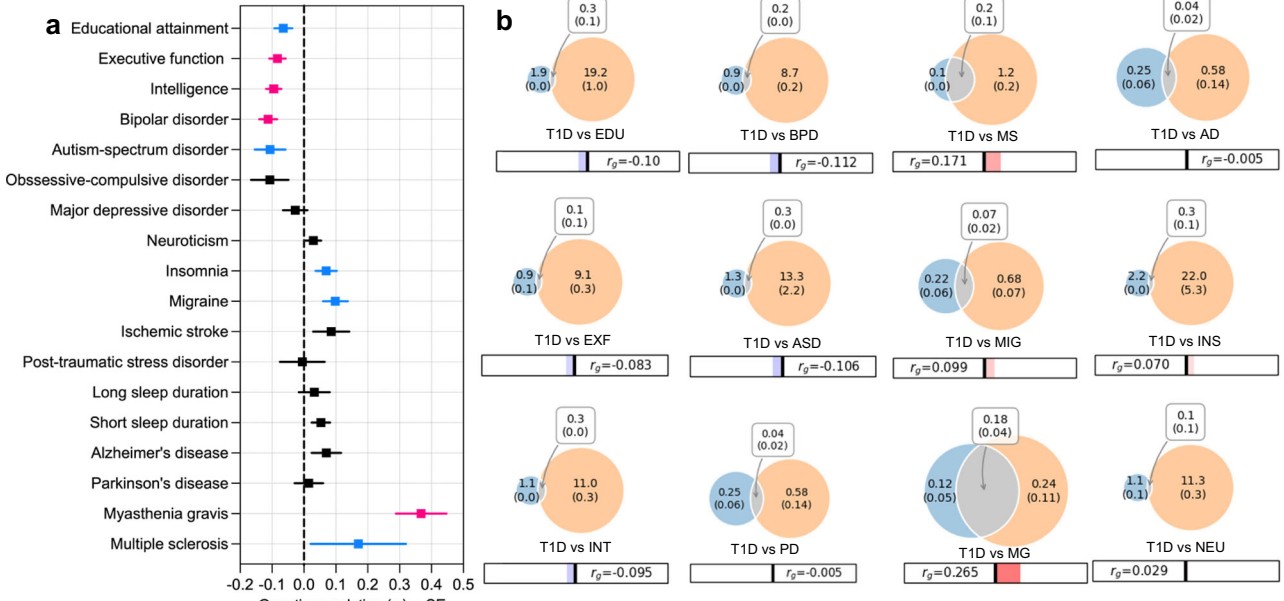

**Fig. 2 | Genetic correlations and polygenic overlap between type 1 diabetes and neurocognitive traits. a** Cross-trait linkage disequilibrium score regression (LDSC) was used to estimate genome-wide genetic correlations (rg). Squares indicate point estimates, and horizontal bars indicate the standard error of rg. Significance was assessed using two-sided tests. Red/pink markers denote correlations significant after Bonferroni correction, whereas blue markers denote nominal associations (two-sided *P* < 0.05 but ≥Bonferroni-corrected threshold). Exact *P* values are provided in Supplementary Table 2. **b** MiXeR bivariate causal mixture models estimating the number of trait-influencing variants for each phenotype (orange) and for type 1 diabetes (blue), with the overlapping area representing shared variants. Numbers inside circles indicate estimated polygenicity (s.e.), and numbers in the overlap indicate the number of shared causal variants. rg values from LDSC are shown below each panel. Statistical analyses were performed using publicly available genome-wide association study summary statistics. The unit of analysis corresponds to individual participants in the original genome-wide association studies. No biological or technical replicates were generated in this study.

overlap co-occurred with weak or inverse genome-wide genetic correlations, indicating that shared variants may exert heterogeneous or opposing effects that are averaged out in LDSC-based correlation estimates.

Taken together, the LDSC and MiXeR analyzes provide complementary perspectives, whereby the LDSC highlights the direction of genome-wide associations, while MiXeR underscores the pervasiveness of shared causal variants irrespective of direction. These support a model in which common brain–immune pathways exert pleiotropic influences across autoimmune, cognitive, and psychiatric outcomes.

### Conjunctional false discovery analysis identifies loci jointly influencing T1D and neurocognitive traits

To resolve locus-level overlap, we applied conjunctional false discovery rate (conjFDR) analysis, which leverages cross-trait enrichment to identify pleiotropic loci beyond those identified using conventional genome-wide significance thresholds[38,39]. Across trait pairs, the number of loci jointly associated with T1D and neurocognitive traits ranged from fewer than 10 to more than 100, with particularly extensive overlap for neuroautoimmune disorders such as multiple sclerosis (109 loci) and myasthenia gravis (69 loci) (Fig. 3a–h, and Supplementary Data 2). Substantial sharing was also evident for cognitive traits, including intelligence (63 loci), educational attainment (45 loci), and EXF (21 loci), and for psychiatric and neurodegenerative conditions such as Alzheimer's disease (29 loci), BPD (13 loci), and Parkinson's disease (9 loci). Importantly, conjFDR pleiotropic locus counts did not show a robust or consistent relationship with GWAS power across LDSC-derived metrics (mean $\chi^2$ and SNP-heritability Z-score; Supplementary Fig. 2, and Supplementary Table 2). Traits spanning a wide range of effective GWAS power exhibited both high and low degrees of pleiotropic overlap with T1D, supporting the interpretation that conjFDR locus counts primarily reflect trait-specific shared genetic architecture rather than GWAS power.

Among the shared loci, the 17q21.31 region emerged as a pleiotropic hotspot, jointly associated with T1D and multiple

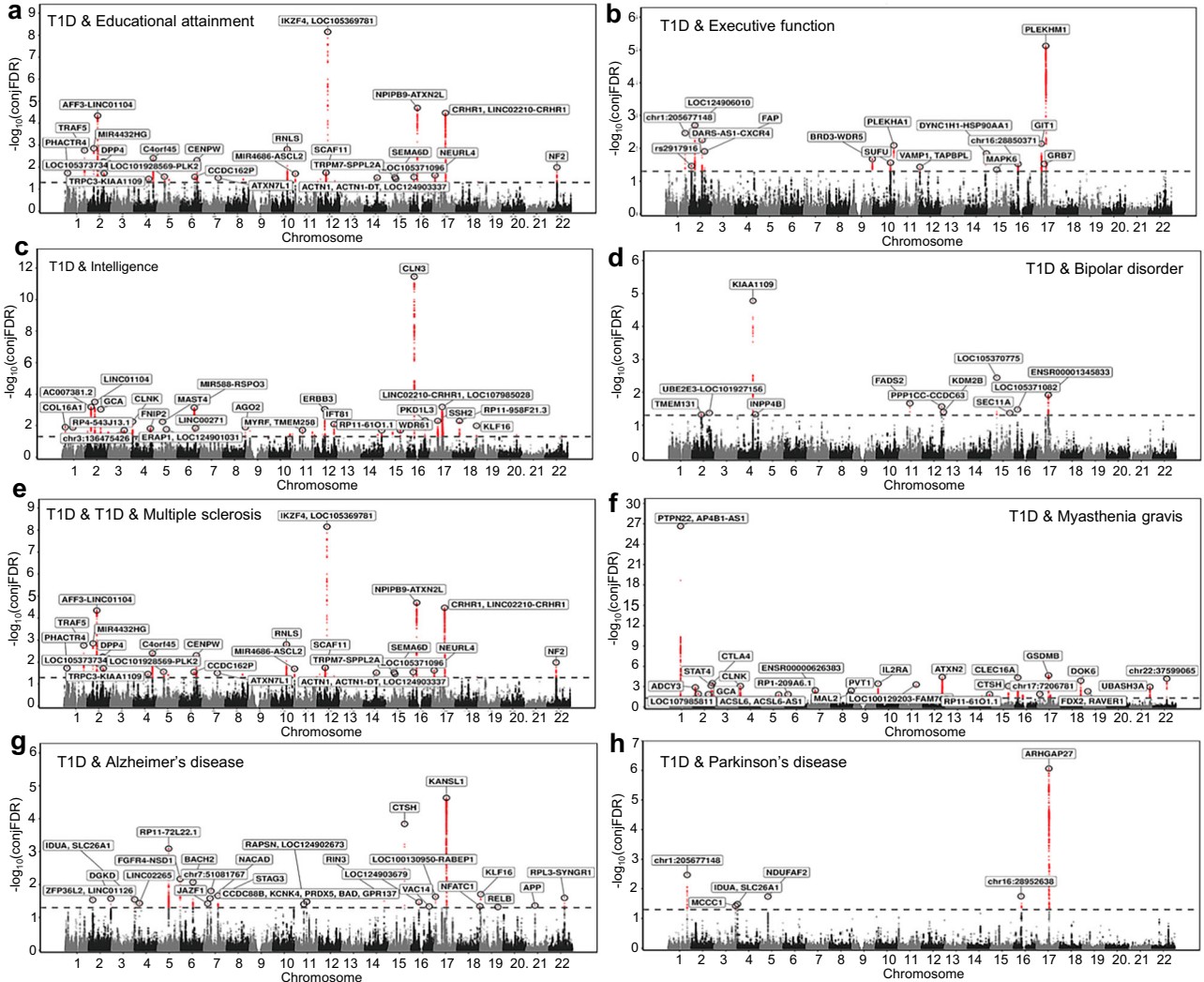

**Fig. 3 | Conjunctional false discovery rate identifies loci jointly associated with type 1 diabetes and neurocognitive traits.** Conjunctional false discovery rate (conjFDR) Manhattan plots showing loci jointly associated with type 1 diabetes and educational attainment (**a**), executive function (**b**), intelligence (**c**), bipolar disorder (**d**), multiple sclerosis (**e**), myasthenia gravis (**f**), Alzheimer's disease (**g**), and Parkinson's disease (**h**). Each point represents a single nucleotide polymorphism plotted by genomic position (*x*-axis) and −log₁₀(conjFDR) value (*y*-axis). Red points denote variants surpassing the conjFDR significance threshold (conjFDR <0.05) and are annotated with the nearest gene. The horizontal dashed line marks the significance threshold (conjFDR = 0.05). ConjFDR integrates genome-wide association study summary statistics from type 1 diabetes and each corresponding trait to identify shared genetic associations beyond single-trait significance. Analyses are based on independent genome-wide association study summary statistics described in Supplementary Table 1. Raw conjFDR results are provided in Supplementary Data 2. The unit of analysis corresponds to individual participants in the original studies, and no biological or technical replicates were generated in this work.

neurocognitive traits. This structurally polymorphic region contains *MAPT*, *KANSL1*, *CRHR1*, and *LRRC37A*, genes implicated in neurodevelopment, neurodegeneration, and immune regulation, highlighting 17q21.31 as a candidate neuroimmune hub. Notably, this locus has been associated previously with cognitive phenotypes and various neuropsychiatric or neurodevelopmental disorders[40–43].

To assess biological context, we examined tissue-specific enrichment of pleiotropic loci. We focused on three representative pairs spanning cognitive, neuroautoimmune, and psychiatric axes using educational attainment, multiple sclerosis, and BPD, respectively (Fig. 4). These neurocognitive traits also showed substantial LDSC genetic correlation and or MiXeR overlap with T1D. Shared loci between T1D and educational attainment were enriched in brain regions including the cerebellum, frontal cortex, and anterior cingulate cortex. Loci jointly influencing T1D and multiple sclerosis were enriched in immune-related tissues such as EBV-transformed lymphocytes, spleen, whole blood, and small intestine. Shared loci between T1D and BPD showed broad enrichment across

all examined brain regions. Notably, enrichment in the pituitary was unique to BPD and stronger than for other traits, suggesting a potential neuroendocrine component to the shared genetic architecture.

These results indicate that pleiotropic loci linking T1D with neurocognitive traits are distributed across biologically distinct and coherent regulatory contexts. Loci shared between T1D and cognitive or psychiatric traits show preferential enrichment in brain tissues, whereas loci shared between T1D and neuroautoimmune traits are enriched in immune tissues, thereby underscoring a dual brain–immune architecture of T1D pleiotropy.

## Bidirectional Mendelian randomization identifies directional and reciprocal associations between T1D and neurocognitive traits

To investigate potential causal relationships (i.e., vertical pleiotropy) between T1D and neurocognitive traits, we conducted bidirectional two-sample Mendelian randomization (MR) analyzes. The inverse-

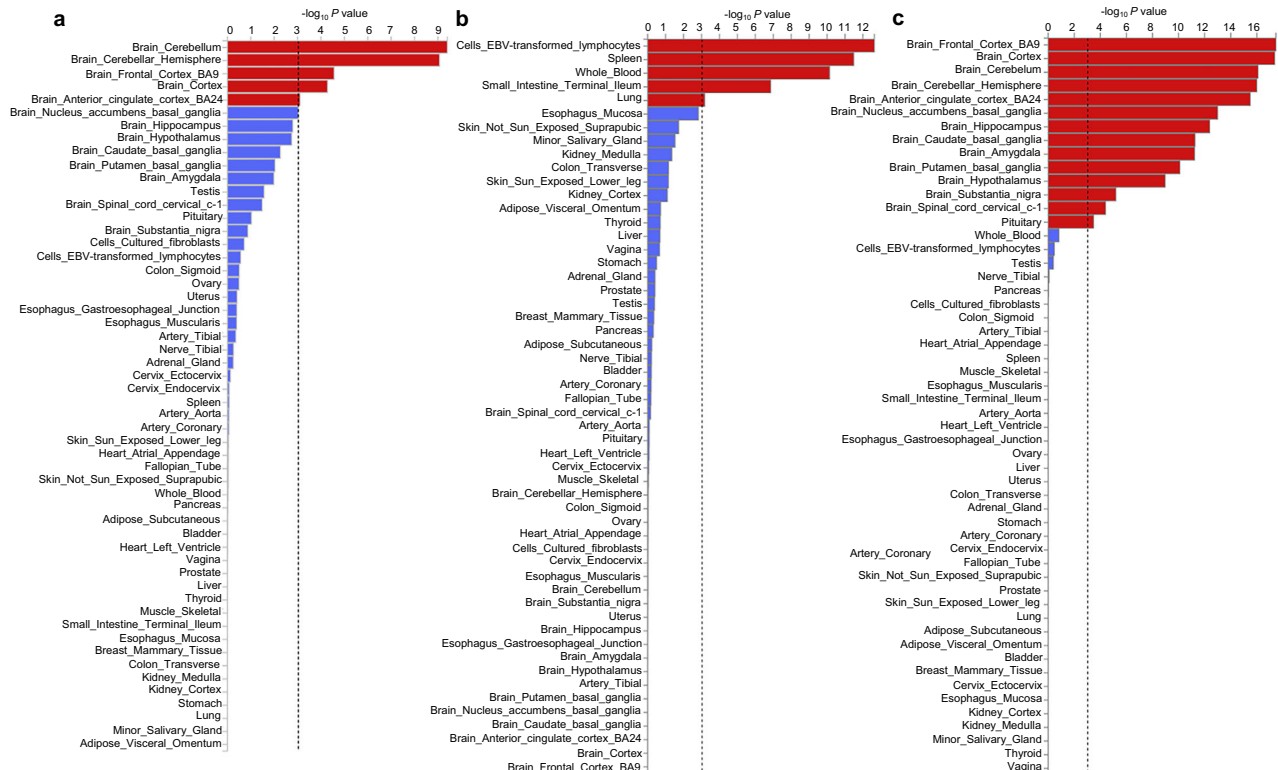

**Fig. 4 | Tissue enrichment of loci jointly associated with type 1 diabetes and neurocognitive traits.** MAGMA tissue enrichment of pleiotropic loci jointly associated with type 1 diabetes and representative neurocognitive traits, including educational attainment (**a**), multiple sclerosis (**b**), and bipolar disorder (**c**). Bars represent $-\log_{10}(P)$ values for enrichment across GTEx tissues. Red bars indicate tissues surpassing nominal significance, whereas blue bars indicate enrichment values that do not reach nominal significance. Significance was assessed using two-sided tests implemented in MAGMA. The vertical dashed line denotes the nominal significance threshold ($P \le 0.05$). Analyses are based on independent genome-wide association study summary statistics described in Supplementary Table 1. Raw conjFDR results used for enrichment analyses are provided in Supplementary Data 2. The unit of analysis corresponds to individual participants in the original studies, and no biological or technical replicates were generated in this work.

variance weighted (IVW) method was used as the primary estimator, with weighted median and MR-Egger models applied as sensitivity analyzes to assess robustness and residual pleiotropy. Bonferroni correction was applied across all IVW tests.

When modeling genetic liability to T1D as the exposure (Fig. 5a, and Supplementary Data 3), we observed a Bonferroni-significant association with increased risk of myasthenia gravis (OR = 1.20, 95% CI: 1.12–1.28, $P = 7.0 \times 10^{-8}$). Several additional associations reached nominal significance ($P < 0.05$) but did not survive Bonferroni correction ($P \ge 2.38 \times 10^{-3}$), including positive associations with migraine and ischemic stroke, and inverse associations with schizophrenia, BPD, Parkinson's disease, and amyotrophic lateral sclerosis (ALS).

In the reverse direction, where genetic liability to neurocognitive traits served as the exposure (Fig. 5b, and Supplementary Data 3), higher liability to multiple sclerosis (OR = 1.18, 95% CI: 1.09–1.27, $P = 2.3 \times 10^{-5}$) and myasthenia gravis (OR = 1.22, 95% CI: 1.12–1.33, $P = 5.6 \times 10^{-6}$) was associated with increased T1D risk. In contrast, higher genetic liability to educational attainment (OR = 0.67, 95% CI: 0.53–0.85, $P = 8.2 \times 10^{-4}$), intelligence (OR = 0.81, 95% CI: 0.71–0.93, $P = 2.0 \times 10^{-3}$), Alzheimer's disease (OR = 0.80, 95% CI: 0.73–0.88, $P = 4.2 \times 10^{-6}$), and BPD (OR = 0.82, 95% CI: 0.76–0.89, $P = 1.2 \times 10^{-6}$) was associated with reduced T1D risk. Genetic liability to obsessive-compulsive disorder, short sleep duration, attention-deficit/hyperactivity disorder, and Parkinson's disease showed nominal associations with T1D risk but did not remain significant after Bonferroni correction.

Collectively, these analyzes reveal both directional and reciprocal genetic associations between T1D and neurocognitive traits, with particularly strong evidence supporting a protective effect of cognitive traits on T1D risk.

### Shared gene expression regulatory variation in brain and immune cells links T1D and neurocognitive traits

We integrated GWAS signals for T1D and neurocognitive traits with expression quantitative trait loci (eQTLs) from bulk brain tissues, single-nucleus brain cell types, and peripheral immune populations using SMR/HEIDI (summary-data Mendelian randomization with heterogeneity in dependent instruments) to identify gene-expression regulatory mechanisms mediating shared genetic liability between T1D and neurocognitive traits. These analyzes revealed pleiotropic eQTLs spanning chromosomes 1, 5, 6, 12, 16, and 17 in brain and peripheral immune compartments that jointly influence T1D risk and neurocognitive traits (Fig. 6, and Supplementary Data 4).

Higher *AP4B1* expression on the 1q25.3 locus in monocytes was associated with reduced risk of both T1D and myasthenia gravis, consistent with its established roles in vesicular trafficking and immune regulation. On 5q11.2, increased *ANKRD55* expression in CD4⁺ T cells conferred elevated risk of both T1D and multiple sclerosis, aligning with previous studies implicating this locus in autoimmune susceptibility. A similar pattern was observed at 6p22.2, where increased *CENPW* expression in stimulated CD8⁺ T cells associated with lower T1D risk and reduced cognitive performance (INT, EDU, EXF).

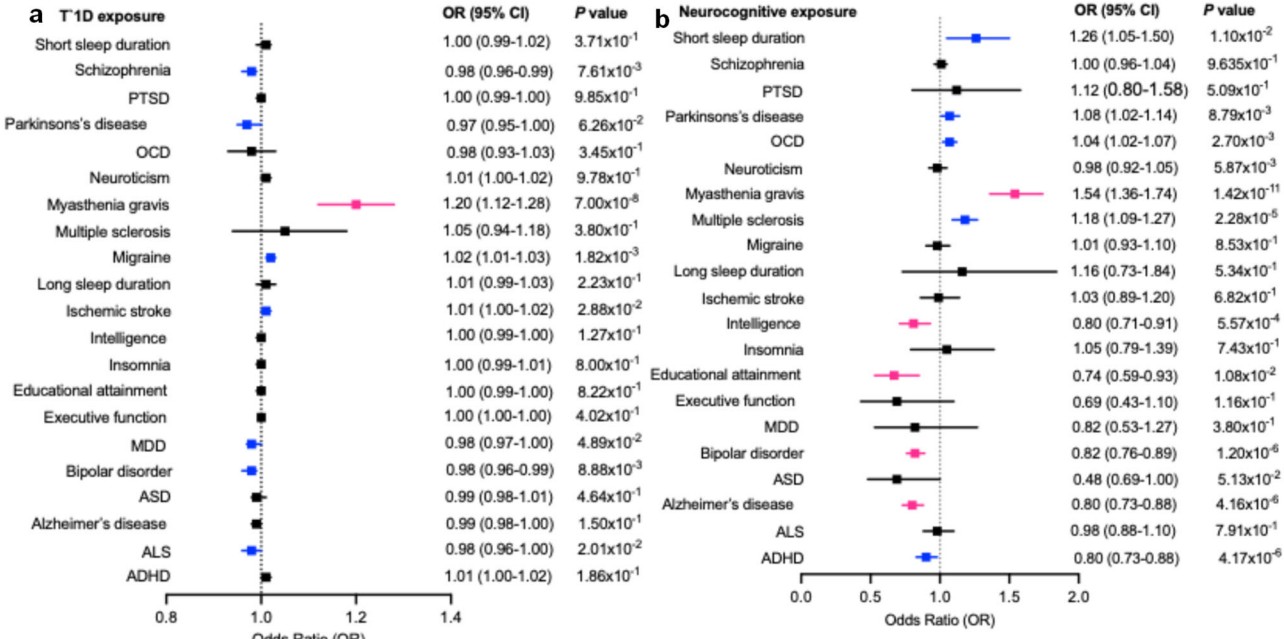

**Fig. 5 | Bidirectional Mendelian randomization (MR) analysis of type 1 diabetes and neurocognitive traits. a** Effects of genetic liability to type 1 diabetes on risk of neurocognitive traits. **b** Effects of genetic liability to neurocognitive traits on risk of type 1 diabetes. Odds ratios (ORs; squares) with 95% confidence intervals (horizontal lines) are shown from inverse variance weighted (IVW) models. Red/pink markers denote Bonferroni-corrected significant associations, while blue markers denote nominal associations (two-sided *P* < 0.05 but ≥0.05/21). Confidence intervals are asymmetric because MR estimates and standard errors are calculated on the log-odds scale and exponentiated to generate odds ratios. Results from weighted median and MR-Egger models are provided in Supplementary Data 3. Analyses are based on independent genome-wide association study summary statistics described in Supplementary Table 1. The unit of analysis corresponds to individual participants in the original genome-wide association studies; no biological or technical replicates were generated in this study.

A richer pleiotropic architecture was observed at 12q13.2, involving *RPS26*, *ERBB3*, and *SUOX*, which exhibited coordinated but directionally distinct effects. Increased *RPS26* expression in brain regions (substantia nigra, hypothalamus) and dendritic cells was associated with higher T1D risk and higher EDU/INT/EXF. By contrast, increased *ERBB3* and *SUOX* expression in T-cell subsets and in the nucleus accumbens for *SUOX* was associated with reduced T1D risk and lower cognitive performance traits. A second regulatory triad was identified at 16p11.2, where increased *NPIPB7* and *SULT1A1* expression conferred elevated risk of both T1D and cognitive traits, whereas increased *TUFM* expression showed the opposite pattern, jointly reducing risk of T1D and EDU/INT/EXF. These directionally coherent effects highlight locus-specific bidirectional pleiotropy in a region known to regulate immunometabolic and neurodevelopmental processes.

The most extensive cluster of pleiotropic signals was observed at 17q21.31, a well-established neurodevelopmental and neuropsychiatric locus[40–43]. Multiple transcripts, including *CRHR1*, *KANSL1*, *ARL17A/B*, *PLEKKHM1*, *LRRC37A2*, *MAPT-AS1*, and others, displayed shared regulatory associations across microglia, inhibitory and excitatory neurons, astrocytes, endothelial cells, and peripheral immune subsets. These transcripts exhibited highly consistent cross-trait patterns in different brain regions and cell types, influencing T1D, EDU/INT/EXF, BPD, PD, autism spectrum disorder, and migraine. Together, these results reinforce 17q21.31 as a major neuroimmune regulatory hub linking autoimmunity, neuropsychiatry, and cognition.

To explore whether genetically implicated regulatory effects from the SMR/HEIDI analyzes correspond to transcriptional changes in human disease-relevant context, we examined single-nucleus RNA-seq and proteomic profiles from the dorsolateral prefrontal cortex of patients with Parkinson's disease (PD) and control individuals. This analysis was motivated by three considerations: (i) several prior MR studies[44,45] and ours (Fig. 6, and Supplementary Data 4) revealed asymmetric bidirectional associations between PD and T1D; (ii) our conjFDR identified 17q21.31 as jointly associated with T1D and multiple neurocognitive traits, with the most significant association being with PD at this locus, and our SMR/HEIDI analyzes further indicated shared regulatory architecture at this locus between T1D and PD along with other neurocognitive traits; and (iii) we had available matched snRNA-seq and proteomic datasets from PD cortex[46], enabling direct in vivo evaluation of genetically inferred effects.

Consistent with the SMR/HEIDI predictions, multiple transcripts at 17q21.31, together with altered protein abundance of *RPS26* at 12q13.2, showed differential expression between PD and control donors in directions concordant with their genetically inferred effects on T1D liability (Supplementary Fig. 3). These findings provide orthogonal evidence that the regulatory programs linking T1D to neurocognitive traits are perturbed in disease-affected human cortex.

Overall, these results demonstrate that pleiotropic gene-expression regulation across 1q25.3, 5q11.2, 6p22.2, 12q13.2, 16p11.2, and 17q21.31 constitutes a mechanistic axis through which shared genetic liability to T1D and neurocognitive traits is mediated across brain and immune cell types.

## Discussion

T1D has long been associated with cognitive alterations and increased psychiatric risk, particularly in individuals with childhood-onset disease[11,17–19], yet it is classically viewed as a peripheral autoimmune condition. In this study, we applied an integrative genomics framework to uncover shared genetic, regulatory, and cellular mechanisms linking T1D with a spectrum of neurocognitive traits across both brain and peripheral immune compartments. These findings support a neuroimmunogenetic model of T1D, in which autoimmunity and altered

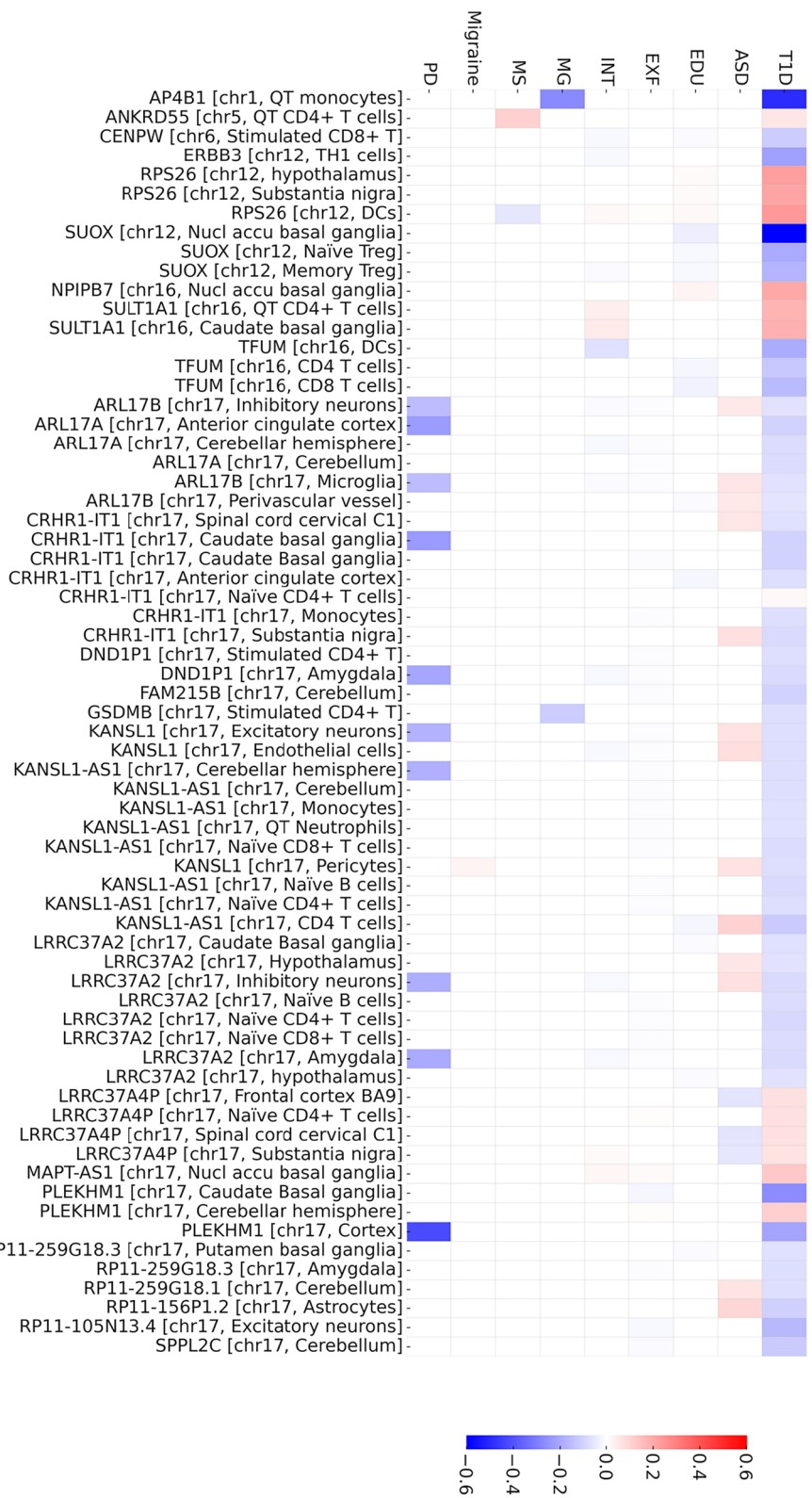

neurocognition reflect shared molecular architecture rather than secondary glycemic effects alone.

Our study reveals that T1D heritability is enriched in accessible chromatin regions of brain-resident cells, particularly microglia, across prenatal and postnatal development into adulthood. Microglia are central players in immune surveillance and synaptic refinement, positioning them at the nexus of neural–immune crosstalk. The enrichment of T1D genetic risk in microglia during early development suggests that genetic alterations of neuroimmune programs may influence both autoimmune susceptibility and neurodevelopmental processes even before T1D onset, potentially contributing to the cognitive alterations frequently observed in patients with early-onset disease.

**Fig. 6 | Shared gene regulation in brain and immune cells links type 1 diabetes with neurocognitive traits.** Heatmap of genes identified by summary data–based Mendelian randomization with HEIDI filtering (SMR/HEIDI), showing expression quantitative trait loci (eQTLs) with pleiotropic associations between type 1 diabetes and at least one neurocognitive trait. Genes are ordered by chromosome, with the tissue or cell type in which the regulatory effect was detected indicated in brackets. Columns represent traits: autism spectrum disorder (ASD), educational attainment (EDU), executive function (EXF), intelligence (INT), myasthenia gravis (MG), multiple sclerosis (MS), migraine, and Parkinson's disease (PD). The color scale represents the direction and magnitude of β effect sizes, with red indicating higher gene expression associated with increased disease risk (or trait liability) and blue indicating higher gene expression associated with reduced risk. Vertical dotted lines denote chromosome boundaries. Only genes passing false discovery rate (FDR)–corrected significance thresholds for SMR and with HEIDI $P > 0.05$ are shown. Source data are provided in Supplementary Data 4. Statistical analyses were performed using publicly available genome-wide association study summary statistics; the unit of analysis corresponds to individual participants in the original studies, and no biological or technical replicates were generated.

The genome-wide genetic correlation and polygenic overlap analyzes demonstrated that T1D shares substantial, and often inverse, genetic architecture with cognitive and psychiatric traits. Specifically, T1D showed negative genetic correlations with educational attainment (EDU), EXF, and intelligence (INT), and MR analyzes supported protective effects of higher EDU and INT liability on T1D risk. These observations suggest potential neurodevelopmental resilience against autoimmunity. Conversely, T1D liability conferred increased risk for neuroautoimmune disorders such as myasthenia gravis, and nominally reducing risk for schizophrenia, BPD, and Parkinson's disease. The inverse association with schizophrenia is consistent with prior epidemiological and MR studies[19]. Although our MR analyzes provided only nominal evidence for relationships between T1D and Parkinson's disease, the asymmetry in effect direction, wherein T1D liability trend toward reduced PD risk while PD liability nominally increased T1D risk, is concordant with prior studies[44,45] and may reflect pleiotropic interactions between dopaminergic neurodegeneration and systemic immune dysregulation[47]. The causal associations of T1D with myasthenia gravis and migraine further underscore the role of neuroimmune crosstalk, as both conditions are increasingly recognized to involve overlapping immune and neuronal regulatory pathways[48,49]. Similar nominal associations were also observed for ALS, consistent with evidence that glial and immune dysfunction contribute to ALS pathogenesis[50], suggesting possible shared mechanisms with neurodegeneration that warrant further investigation.

At first glance, the negative genetic associations of T1D with schizophrenia and BPD, alongside its negative associations with cognitive traits, may appear contradictory given the well-established negative genetic correlations between BPD/SCZ and cognitive traits. However, these patterns likely arise from distinct biological pathways. Large-scale psychiatric genetics studies show that BPD and SCZ are genetic correlated and their genetic liability is driven primarily by synaptic and neuronal excitation–inhibition regulatory mechanisms[51]. By contrast, our S-LDSC and SCAVENGE analyzes suggest that the T1D–cognition genetic relationship is rooted in neuroimmune pathways, wherein T1D heritability showed strong enrichment in microglia across developmental stages, with additional but developmentally restricted enrichment in early postnatal inhibitory neurons. Thus, the negative T1D–cognition association is unlikely to arise through the same cellular mechanisms that link BPD and SCZ negatively with cognitive impairment. Instead, T1D may influence cognitive and psychiatric liability through immune–glial regulatory programs that differ from the synaptic and neuronal excitation–inhibition regulatory processes underlying BPD and SCZ. This mechanistic distinction could explain why T1D shows negative genetic associations with both cognition and BPD/SCZ, despite the well-established negative correlations of BPD/SCZ with cognitive performance. The bidirectional inverse causal associations between T1D and BPD further illustrate how reciprocal genetic influences can shape comorbidity, risk antagonism, or protective divergence across these disorders.

Mechanistically, we identified several pleiotropic loci converging on shared regulatory axes across chromosomes 1, 5, 6, 12, 16, and 17 in brain and immune cells that jointly shape liability to T1D and neurocognitive traits. Increased *AP4B1* expression at 1q25.3 in monocytes was associated with reduced risk of both T1D and myasthenia gravis, consistent with its established roles in vesicular trafficking and immune regulation. Likewise, increased *ANKRD55* expression at 5q11.2 in CD4⁺ T cells was associated with increased risk of T1D and multiple sclerosis, in line with its well-established autoimmune associations[52,53]. At 6p22.2, higher *CENPW* expression in stimulated CD8⁺ T cells was associated with reduced T1D risk and reduced intelligence. Although *CENPW* has been studied primarily in cell-cycle and chromatin-assembly pathways, these convergent effects suggest shared regulatory influences on immune activation and neurodevelopment.

A pleiotropic triad at the 12q13.2 locus, in which *RPS26*, *ERBB3*, and *SUOX* expression showed coordinated but directionally distinct associations with T1D and neurocognitive traits was observed. Although *RPS26* and *ERBB3* are frequently highlighted as candidate effector genes for T1D at this locus, *SUOX* has not typically been prioritized in autoimmune fine-mapping. Nonetheless, emerging evidence suggests roles for *SUOX* in immunometabolic regulation of T-cell states, including regulatory T cells, and in neurometabolic pathways within the nucleus accumbens involved in affective and reward processing. These provide biological context for its dual immune–brain involvement in T1D and neurocognitive traits. A similar regulatory triad was observed at 16p11.2, where increased *NPIPB7* and *SULT1A1* expression increased risk of both T1D and neurocognitive traits, while increased *TUFM* expression conferred reduced risk of both. These directionally consistent effects again point to bi-directional pleiotropy within a locus central to immune-metabolic and neurodevelopmental regulation.

The most prominent neuroimmune locus was 17q21.31, where multiple transcripts, including *CRHR1, KANSL1, ARL17A, ARL17B, PLEKHM1, LRRC37A2,* and flanking genes, showed shared regulatory effects across immune and brain-resident cells, influencing T1D, cognitive, and psychiatric phenotypes. This locus has long been associated with synaptic plasticity, stress response, and neurodevelopment[42,43] and has been implicated in several neuropsychiatric disorders, including schizophrenia, BPD, autism, Parkinson's disease, and depression[40–43]. Notably, the region exhibited highly coherent cross-trait directions of effect, underscoring its role as a major point of convergence for immune and neural pathways and supporting a previously unrecognized neuroimmune contribution of 17q21.31 to T1D pathogenesis.

Consistent with these SMR/HEIDI predictions, implicated genes showed concordant patient–control expression differences in single-nucleus RNA-seq and proteomic datasets from Parkinson's disease, a disorder that both our analyzes and prior studies have shown to causally influence T1D risk[44,45]. Collectively, these findings support pleiotropic transcriptional regulation across 12q13.2, 16p11.2, and 17q21.31, as well as additional hubs on chromosomes 1, 5, and 6, as integrative neuroimmune mechanisms linking T1D and neurocognitive traits.

Although LDSC analyzes revealed negative genetic correlations between T1D and neurocognitive traits, expression-based colocalization (SMR/HEIDI) analyzes added a more nuanced perspective. Specifically, most eQTLs with pleiotropic effects on T1D and cognitive traits (EDU, INT, EXF) showed concordant expression-mediated effects, such that increased gene expression was associated with both higher

cognitive performance and elevated T1D risk. At first glance, this appears discordant with the negative genome-wide genetic correlations and the protective MR effects of cognitive traits (EDU, INT) on T1D. However, this discrepancy reflects fundamental differences in analytic resolution, as LDSC captures average direction of genome-wide polygenic overlap[54], whereas SMR/HEIDI isolates locus-specific, expression-mediated pleiotropy[55]. Although MR supports a causal effect of cognitive traits on reduced T1D risk, this causal pathway likely involves mechanisms not fully captured by steady-state eQTL-mediated regulation. Overall, these patterns underscore that genome-wide polygenic relationships, causal trait effects, and regulatory mechanisms can diverge in both direction and tissue specificity.

The results of this study redefine our understanding of T1D by situating it within a broader neurodevelopmental and neuroimmune context. Rather than an isolated autoimmune pathology, T1D emerges as a pleiotropic trait whose genetic architecture converges with that of cognition, psychiatric traits, and neuroinflammation. This convergence is rooted in shared regulatory variation, developmentally dynamic brain–immune interactions, and pleiotropic gene expression programs spanning both central and peripheral systems.

This study also motivates deeper mechanistic investigation of microglial and glial contributions to T1D pathogenesis, particularly during early neurodevelopment. The study also highlights potential biomarkers and therapeutic targets, such as *CRHR1, LRRC37A2, KANSL1* and *RPS26* that may modulate both immune and neurocognitive outcomes. The observed neuroimmune convergence offers preliminary translational insight, pointing to microglial and T-cell–linked regulatory programs as candidates for future mechanistic and therapeutic exploration. Although speculative, these findings provide a genomic framework for prioritizing targets relevant to both autoimmunity and neurocognitive function. Third, the work underscores the utility of integrative genomics in resolving complex pleiotropy across traits traditionally viewed as organ-specific. Finally, it suggests that cognitive alterations and the observed increases or decreases in neuropsychiatric disease risk in T1D may not be mere consequences of dysglycemia but instead reflect shared genetic origins.

This study is not without limitations. Although two-sample MR provides directional inference, some degree of participant overlaps between exposure and outcome GWAS, particularly for traits incorporating UK Biobank, remains possible. However, the bivariate LDSC intercepts for all trait pairs were close to 1.0, indicating minimal residual covariance and suggesting that any overlap is unlikely to materially affect the Bonferroni-significant MR associations. MR estimates with effect sizes near the null may be more sensitive to even small degrees of overlap, and these should therefore be interpreted cautiously. Future work applying overlap-robust MR approaches, such as MRBEE[56], may help further validate these causal relationships. In addition, because age at T1D diagnosis is not consistently available across cohorts in the underlying GWAS meta-analysis, future work using age-of-onset–stratified T1D GWAS will be important to determine whether the neurocognitive associations differ between childhood- and adult-onset T1D. Furthermore, while we leveraged diverse eQTL resources, tissue- and context-specific regulatory dynamics remain incompletely captured, particularly in disease-relevant states. Future work integrating perturbation-based single-cell models and longitudinal cohorts will be critical to functionally validate these findings. Finally, while our analyzes focused on individuals of predominantly European ancestry, the extent to which these neuroimmune mechanisms generalize across global populations remains to be determined.

In summary, this study provides genomic evidence that T1D is embedded within a broader neurocognitive and neuroimmune architecture. The convergence of genetic, epigenomic, and transcriptomic data across brain and immune compartments highlights novel pathways linking autoimmunity with cognitive function and mental health. Consistent with prior studies demonstrating that autoimmunity constitutes a modifiable risk component for dementia[57], our findings extend these observations to a wider cognitive and psychiatric context and provide high-resolution, cell-type-specific evidence, particularly implicating microglia and inhibitory neurons as neuroimmune convergence points of T1D genetic liability. These insights offer a unified conceptual framework for understanding comorbidity and suggest that targeting neuroimmune interfaces may benefit both metabolic and cognitive outcomes, supporting the rationale for translational strategies that incorporate immunomodulation in the prevention of autoimmune and neurodegenerative diseases.

## Methods

### Study datasets and quality control
We used GWAS summary datasets and eQTL resources derived from individuals of predominantly European ancestry to minimize population heterogeneity and align with European linkage disequilibrium (LD) reference panels. The primary GWAS for T1D comprised 520,580 individuals (18,942 cases). Additional GWAS included cognitive traits (intelligence, educational attainment, EXF), psychiatric disorders (autism spectrum disorder, BPD, schizophrenia, obsessive-compulsive disorder, ADHD, major depression, PTSD, and neuroticism), neurological and neurodegenerative diseases (Alzheimer's disease, Parkinson's disease, ALS, multiple sclerosis, myasthenia gravis, migraine, ischemic stroke), and sleep-related phenotypes (insomnia and sleep duration).

For regulatory analyzes, we incorporated multiple eQTL datasets: bulk brain tissue from GTEx v8 ($N = 838$), single-cell brain eQTLs spanning eight major cell types ($N = 192$), single-cell blood eQTLs ($N = 982$), and immune-cell eQTLs across 18 purified cell populations ($N = 200$). These datasets enabled assessment of genetic regulation across both central nervous system and immune compartments.

Across all analyzes, we applied stringent quality control procedures. SNPs were restricted to autosomal variants with minor allele frequency (MAF) > 1% and imputation INFO > 0.9. Variants with ambiguous strands were removed, and the extended major histocompatibility complex (MHC) region (chr6:25–34 Mb) was excluded from analyzes requiring LD independence due to complex long-range LD. Allele harmonization was performed across all datasets. LD estimation was based on the European reference panel from the 1000 Genomes Project Phase 3, ensuring ancestry-matched analyzes.

This research complies with all the ethical regulations related to the secondary analysis of data collected by various cohorts, each of which obtained ethical approval and informed consent. All datasets used for analyzes are summarized in Supplementary Table 1.

### Stratified LD score regression (S-LDSC)
We applied stratified LD score regression (S-LDSC)[31] to estimate whether T1D SNP-heritability was enriched in accessible chromatin regions of specific cortical cell types across human neurodevelopment. Single-nucleus ATAC-seq (snATAC-seq) peak annotations were obtained from human cortex spanning prenatal, early postnatal (0–4 years), late postnatal (4–20 years), and adulthood (>20 years) developmental stages[33]. Peaks were converted into binary annotation files for each cell type and developmental stage.

S-LDSC analyzes were performed using S-LDSC v1.0.1 with the baseline-LD model v2.2 and European reference LD scores from the 1000 Genomes Project Phase 3. Heritability enrichment was quantified as the proportion of SNP-heritability explained by each annotation divided by the proportion of SNPs overlapping that annotation. Statistical uncertainty was assessed using standard errors provided by the S-LDSC framework, and nominal *p*-values are reported to summarize evidence for enrichment. Results are interpreted as patterns of heritability enrichment across annotations.

**SCAVENGE analysis.** SCAVENGE (Single Cell Analysis of Variant Enrichment through Network propagation of Genomic data) is a computational algorithm that uses network propagation to map causal variants to their relevant cellular context at single-cell resolution[36]. SCAVENGE mitigates sparsity in single-cell epigenomic data by leveraging network propagation: a small subset of "seed" cells enriched for trait-relevant variants are identified, and trait information is propagated across a nearest-neighbor cell graph to generate TRS for all cells. Following published procedures[36,37], we implemented SCAVENGE analysis to integrate fine-mapped genetic variants for T1D ($n = 520,580$ samples)[32] with single-nucleus ATAC-seq (snATAC-seq) profiles of human cortex across distinct developmental stages—prenatal, early postnatal (0–4 years), late postnatal (4–20 years), and adulthood (>20 years)[33]. This analysis yielded TRS that quantify the contribution of each cell type and stage to T1D genetic predisposition.

To initiate propagation, fine-mapped T1D variants were intersected with accessible chromatin peaks to identify seed cells. Trait information from seed cells was propagated across a cell-to-cell similarity network, generating TRS for all nuclei. Cells with TRS > 2 were classified as enriched, consistent with the SCAVENGE framework[36]. To validate enrichment at the population level, TRS were aggregated across annotated cell types and developmental stages, and statistical significance was assessed using hypergeometric testing with false discovery rate (FDR) correction[37].

For benchmarking and cross-trait contextualization, the same workflow was applied using genome-wide significant loci from Alzheimer's disease[58], BPD[59] and height[60] GWAS, enabling direct comparisons of developmental and cell-type-specific enrichment patterns across disorders. All SCAVENGE analyzes were performed in R using the SCAVENGE software (v.1.0.2).

## Genome-wide genetic correlation analysis by cross-trait LDSC

We applied cross-trait LDSC to estimate genome-wide genetic correlations[61] between T1D and each neurocognitive trait. Narrow-sense SNP-heritability and pairwise genetic correlations were computed from GWAS summary statistics for T1D and each neurocognitive phenotype. As previously described[61], LDSC estimates genetic correlation as the proportion of shared genetic variance between traits attributable to common SNP effects across the genome. Analyzes were performed using LDSC v1.0.1 with pre-computed LD scores from European populations of the 1000 Genomes Project Phase 3. SNPs with MAF > 0.05 were retained, while variants in the major histocompatibility complex (MHC, chr6:25–34 Mb) region were excluded due to extensive long-range LD.

We corrected for multiple testing using the Bonferroni method, applying a significance threshold of $P < 0.00238$ (0.05/21 trait pairs). We also applied the Benjamini–Hochberg false-discovery rate (BH–FDR) procedure across all tests, considering $q < 0.05$ statistically significant, while associations with $P < 0.05$ but $q \geq 0.05$ are reported as suggestive. Both FDR-adjusted $q$-values and Bonferroni significance indicators are reported in Supplementary Data 1.

## Quantification of polygenic overlap by MiXeR analysis

We applied MiXeR[62] to quantify polygenic overlap between T1D and neurocognitive traits, providing a more nuanced measure of shared genetic architecture beyond global genetic correlation. MiXeR implements Gaussian causal mixture models to estimate both the number of causal variants unique to each trait and the number of variants with pleiotropic effects.

**Univariate modeling.** We first performed univariate MiXeR analysis (MiXeR v1.3) for T1D and each neurocognitive trait to estimate trait polygenicity (the number of SNPs accounting for 90% of SNP-heritability, $h^2_{SNP}$) and discoverability (the average effect size of trait-

influencing variants). Variants with negligible effect sizes were excluded to ensure robust estimates.

**Bivariate modeling.** We then fit bivariate models to each T1D-neurocognitive trait pair to estimate the number of shared and unique causal variants. Model fit and convergence were evaluated using likelihood profiles and residual inspection. Dice coefficients were computed to summarize polygenic overlap on a 0–100% scale[63] and estimates of genome-wide genetic correlation were obtained using cross-trait LDSC for comparison.

**Visualization.** Results were summarized in Venn diagrams showing the number of unique and shared variants per trait pair. Conditional quantile–quantile (Q–Q) plots were generated to assess enrichment of SNP associations across traits[64]. Q–Q plots display the empirical cumulative distribution of $P$ values in one phenotype, stratified by significance thresholds in the second phenotype (e.g., $P \leq 0.1$, $P \leq 0.01$, $P \leq 0.001$), with leftward deflection from the null line indicating cross-trait enrichment.

**Quality control.** Analyzes were restricted to autosomal SNPs with MAF > 0.01 and imputation INFO score >0.9, after removing strand-ambiguous and multi-allelic variants. The MHC region was excluded due to long-range linkage disequilibrium (LD). To ensure consistent LD structure across datasets, we used the European subset of the 1000 Genomes Project Phase 3 reference panel ($n = 503$) for LD estimation. All summary statistics were harmonized to the same genome build (GRCh37/hg19) with allele alignment performed against the reference panel.

The MiXeR analyzes were conducted in R (v4.x) using the MiXeR v1.3 package with default settings unless otherwise specified.

## Shared locus and variant discovery using conjunctional FDR (conjFDR)

We applied conjFDR analysis[38,39] to identify shared genetic loci influencing T1D and each neurocognitive phenotype. ConjFDR leverages cross-trait GWAS summary statistics to increase power for detecting pleiotropic variants while accounting for polygenic overlap.

As a first step, conditional quantile–quantile (Q–Q) plots were generated to evaluate enrichment of T1D-associated variants as a function of their significance in each neurocognitive phenotype (and vice versa). Evidence of leftward deflection in conditional Q–Q plots indicates polygenic overlap between the traits. Conditional FDR (condFDR) values were then computed for each SNP, representing the FDR of T1D associations given their significance in the neurocognitive phenotype (and vice versa). The conjFDR statistic was defined as the maximum of the two condFDR values per SNP, providing a conservative estimate of the probability that a variant is null for either or both traits. SNPs with conjFDR <0.05 were considered jointly associated with T1D and the neurocognitive phenotype. Lead variants were annotated as the SNP with the lowest conjFDR value within each LD-independent locus, defined as $r^2 < 0.1$ within a ±250 kb window. LD was estimated using European reference data from the 1000 Genomes Project Phase 3. To minimize spurious associations, we used GWAS summary statistics pre-corrected for relevant covariates and population stratification. Analyzes were restricted to autosomal SNPs, and only a single representative signal was retained in the extended MHC region (chr6:26–34 Mb, hg19) due to complex LD.

All conjFDR analyzes were conducted using the *pleioFDR* R package[65,66] following established workflows.

## Tissue enrichment analysis of shared loci between T1D and neurocognitive traits

We performed tissue enrichment analysis of loci jointly associated with T1D and neurocognitive traits as identified by conjFDR. SNPs with

conjFDR <0.05 were submitted to the Functional Mapping and Annotation (FUMA) platform (v1.5.2)[64] to define LD-independent loci. Independent significant SNPs were identified using the European 1000 Genomes Project Phase 3 reference panel[67], applying thresholds of $r^2 < 0.6$ for clumping and a maximum merging distance of ≤250 kb. Lead SNPs were defined as variants with $r^2 < 0.1$ within each locus, with the lead SNP taken as the variant with the lowest conjFDR value.

Novelty of lead SNPs was assessed by examining whether they achieved genome-wide significance ($P < 5 \times 10^{-8}$) in the original GWAS for T1D or the neurocognitive trait. Genes were assigned to loci by proximity, defined as the gene with the closest transcription start site to the lead SNP. Gene-level analyzes were conducted using MAGMA within the FUMA framework, testing for tissue-specific expression and gene-set enrichment based on GTEx v8 transcriptomic data. Gene expression was quantified in transcripts per million (TPM). Multiple-testing correction was applied using FDR < 0.05. To further prioritize functional candidate genes, we integrated locus-to-gene assignments from Open Targets Genetics[68], focusing on lead SNPs outside of the extended MHC region due to its complex LD.

### Mendelian randomization analyses
We performed bidirectional two-sample Mendelian randomization (MR) to estimate causal relationships between T1D and neurocognitive phenotypes. MR uses genetic variants strongly associated with an exposure as instrumental variables (IVs) to test for potential causal effects of the exposure on the outcome.

**Instrument selection and harmonization.** SNPs associated with each exposure at genome-wide significance ($P < 5 \times 10^{-8}$; F statistic ≥10) were selected as IVs and pruned using LD clumping at $r^2 < 0.001$. Exposure and outcome summary statistics were harmonized by aligning alleles to the forward strand; ambiguous SNPs with non-inferable strands were excluded.

**Primary MR analyses.** SNP-specific causal effects were estimated using the Wald ratio and meta-analyzed across variants with the IVW method under a multiplicative random-effects model[69]. IVW was used as the primary MR test, under the assumption that all IVs are valid and horizontal pleiotropy is absent.

**Sensitivity analyses.** To evaluate robustness to horizontal pleiotropy, we performed MR using the weighted median (WM) and MR-Egger regression methods[69–72]. The weighted median estimator provides consistent causal effect estimates if at least 50% of the weight comes from valid IVs, even if the remaining IVs are invalid. MR-Egger allows for directional pleiotropy by including an intercept term in the regression of outcome coefficients on exposure coefficients; the intercept test was used to evaluate the presence of unbalanced pleiotropy[71–73]. Under the null of no pleiotropy, the MR-Egger intercept approaches zero, and MR-Egger converges to IVW[69,74].

**Multiple testing correction.** We corrected for multiple testing using the Bonferroni method, applying a significance threshold of $P < 0.00238$ (0.05/21 trait pairs). We also applied the Benjamini–Hochberg false-discovery rate (BH–FDR) procedure across all tests, considering $q < 0.05$ statistically significant, while associations with $P < 0.05$ but $q \geq 0.05$ are reported as suggestive. Both FDR-adjusted $q$-values and Bonferroni significance indicators are reported in Supplementary Data 3. Weighted median and MR-Egger estimators were treated as sensitivity analyzes and were not subjected to additional multiplicity correction.

**Reporting.** For each exposure–outcome pair, we report the IVW causal effect estimate as an odds ratio (OR) with 95% confidence intervals (CI), together with the raw $p$-value, BH–FDR $q$-value, and Bonferroni significance flag, representing the change in outcome per one standard deviation (SD) increase in the exposure.

All analyzes were conducted in R (v4.3.2; R Foundation for Statistical Computing, Vienna, Austria) using the *TwoSampleMR* package. Methods and reporting adhered to the STROBE-MR best-practice guidelines[75].

### Summary-based Mendelian randomization and heterogeneity in dependent instruments (SMR/HEIDI) analysis
SMR extends the concept of MR, allowing testing of the hypothesis that genetically determined expression levels of a gene are associated with a phenotype. A key assumption of this approach is that the same underlying causal variant determines both gene expression and the disease/trait. SMR cannot distinguish between vertical pleiotropy—the situation in which variant influences phenotype via gene expression, and horizontal pleiotropy, the situation in which variant influences phenotype and gene expression but influences the phenotype at least partly independently of gene expression. To distinguish pleiotropy from linkage, the HEIDI test was developed[55], which exploits the observation that if gene expression and disease/trait are in vertical pleiotropy with the same causal variant, the causal effect is identical for any variant in linkage disequilibrium (LD) with the causal variant. Thus, greater heterogeneity among effect size statistics calculated for all significant cis-eQTLs implies a greater likelihood that linkage, rather than causality/vertical pleiotropy, explains the observed causal association. The heterogeneity statistic, the "HEIDI" statistic, tests the hypothesis HEIDI = 0. This provides a formal test of heterogeneity, with $P$-values < 0.05 suggestive of linkage, rather than pleiotropy, as the underlying biological model[55]. Thus, the SMR & HEIDI approach is widely used to test if a transcript and phenotype are associated because of a shared causal variant (i.e., pleiotropy). We used SMR/HEIDI to integrate GWAS association signals for T1D and each neurocognitive trait with eQTL derived from bulk brain regions[76], and single-nucleus brain cells[77] and single-cell blood[78] and immune cells[79] to identify genes whose expression levels are jointly associated with T1D and neurocognitive traits. We performed SMR using the SMR software tool (SMR v1.0.2) in the command line using default options[55], i.e., cis-eQTLs selected based on minimum $P = 5 \times 10^{-8}$, eQTLs included for the HEIDI test based on minimum $P = 1.57 \times 10^{-3}$, eQTLs included for the HEIDI test if $R^2$ with the top cis-eQTL was between 0.05 and 0.9, minimum number of SNPs included in the HEIDI test = 3, maximum number of SNPs included in the HEIDI test = 20 and physical window around probe within which the top cis-eQTL was selected = 2 MB. $P$-values were adjusted in R (v3.6.1) to control the FDR at $\alpha = 0.05$ using the Benjamini–Hochberg procedure. Associations with $P_{HEIDI} < 0.05$–cutoff [80]—were considered likely due to linkage. Probes were excluded if any of the transcript or the top eQTL resided within the super-extended MHC (hg19 6:25,000,000–35,000,000) given the complex linkage disequilibrium structures within this region. Linkage disequilibrium estimation was performed using reference genomes obtained from the 1000 genomes samples of European ancestry[81].

### Transcriptomic and proteomic analysis for differential expression of SMR/HEIDI-identified genes in Parkinson's disease-affected and control brain tissue
To investigate whether genetically predicted expression of genes in the brain associated with T1D manifests in human disease contexts, we analyzed our previously published single-nucleus transcriptomic and proteomic data from brain tissue of individuals with Parkinson's disease (PD; $n = 6$) and matched control individuals ($n = 6$)[46]. This focus was motivated by suggestive Mendelian randomization (MR) associations between T1D and PD—particularly the nominal evidence for increased T1D risk with PD liability—together with conjFDR and SMR/HEIDI analyzes identifying pleiotropic eQTLs shared between the two traits. We interrogated our previously published single-nucleus RNA-

sequencing (snRNA-seq) and label-free quantitative proteomics data-sets generated from the prefrontal cortex of PD patients and control subjects[46]. Genes were selected for transcriptomic and proteomic interrogation if their genetically predicted expression in bulk brain tissue (cortex, frontal cortex BA9, substantia nigra) or microglia was significantly associated with T1D risk based on SMR/HEIDI analysis.

Transcript-level differential expression analyzes were conducted using the DESeq2 framework, incorporating covariate adjustment for age, sex, and postmortem interval, and following the analytical pipe-line detailed in our previous publication. Genes with an adjusted $P < 0.05$ were considered differentially expressed.

In parallel, we assessed differential protein abundance of SMR-identified genes using label-free mass spectrometry-based proteomics data from the same cohort. Protein-level comparisons were performed using moderated $t$-tests with Benjamini–Hochberg FDR correction. Given the modest sample size and resulting limited statistical power, we adopted a relaxed significance threshold of $P < 0.1$ to capture a broader set of biologically relevant signals.

### Reporting summary
Further information on research design is available in the Nature Portfolio Reporting Summary linked to this article.

## Data availability
Genome-wide association study (GWAS) summary statistics used in this study are publicly available from the original consortia and repo-sitories listed in Supplementary Table 1, with accession links and/or PMIDs provided. Single-cell ATAC-seq and RNA-seq data are available from the referenced studies as indicated in the Methods. Processed data generated in this work, including results from LDSC, MiXeR, conjunctional false discovery rate, SMR/HEIDI, and Mendelian rando-mization analyses, are provided in the Supplementary Tables. Source data are provided with this paper.

## Code availability
Code for this study is available at GitHub (https://github.com/AlagsLabTeam/T1D-NEURO) and archived at Zenodo[82].

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

## Acknowledgements

This study was supported by Breakthrough T1D grants to DAA (Grant Keys: 5-CDA-2025-1682-S-B and SRA-2024-1472-S-B).

## Author contributions

P.S. and Z.A.S. contributed to the conception and initiation of the project. P.S., Z.A.S., Z.X., Y.D., and B.Z. performed data analyses. D.A.A. conceived, designed, and supervised the study and drafted the manuscript. All authors (P.S., Z.A.S., Z.X., Y.D., A.J., M.S., S.R., B.Z., L.Z., A.T.D., S.A., and D.A.A.) contributed to data interpretation, manuscript review, and approval of the final draft.

## Competing interests

The authors declare no competing interests.

## Additional information

¹Yale Center for Molecular & Systems Metabolism, Yale University School of Medicine, New Haven, CT, USA. ²Department of Comparative Medicine, Yale University School of Medicine, New Haven, CT, USA. ³Program of Computational Biology and Bioinformatics, Yale University, New Haven, CT 06510, USA. ⁴Department of Neurology, Yale University School of Medicine, New Haven, CT, USA. ⁵Department of Neuroscience, Yale University School of Medicine, New Haven, CT, USA. ⁶Department of Chronic Disease Epidemiology, Yale School of Public Health, New Haven, CT, USA. ⁷Center for Perinatal, Pediatric and Environmental Epidemiology, Yale School of Public Health, New Haven, CT, USA. ⁸Institute for Genomic Health, Icahn School of Medicine at Mount Sinai, New York, NY, USA. ⁹Department of Genetics and Genomic Sciences, Icahn School of Medicine at Mount Sinai, New York, NY, USA. ¹⁰These authors contributed equally: Priscilla Saarah, Zehra A. Syeda. ✉e-mail: david.alagpulinsa@yale.edu

