## [Transparent Peer Review file · Nature Communications]

Shared genetic and neuroimmune architecture links type 1 diabetes with neurocognitive traits

Corresponding Author: Dr David Alagpulinsa

Version 0:

Reviewer comments:

Reviewer #1

(Remarks to the Author)

Overview

Motivated by prior demonstrating phenotypic neurocognitive and psychiatric associations, this manuscript provides a series of analyses to evaluate how T1D genetically relates to traits and disorders in these classes of phenotypes at multiple biological levels. The results are comprehensive and give insight into some of the genetic mechanisms that may link T1D to traits like bipolar disorder and educational attainment. The code is also very well documented on a dedicated and public project GitHub page. My primary concern is that the results currently include many lists of findings, but I think this needs to be more effectively distilled down to the main takeaways by doing things like moving nominal results, small effects at global levels, and follow-up results on small global effects to the online supplement. In addition, I think results at more granular levels need to be more carefully interpreted relative to highly polygenic and pleiotropic signal for any human complex trait, including T1D. As currently written, I think the reader would both struggle to take away the most important take home points and, for non-genetics readers, would be tempted to overinterpret single loci (e.g., 17q21.31) or pathways (e.g., microglia) as necessarily mediating the majority of the T1D and neurocognitive genetic overlap.

Major Comments.

1. I strongly recommend cutting the nominal associations from Results and Discussion in favor of simply referring interested readers to supplementary materials to evaluate those outside of the main text. In support of this, some of the nominal MR effects are quite small (e.g., OR = .98) but with small confidence intervals (.96-1), indicating the effect is likely small enough in the population to be near meaningless or entirely accounted for by some genetically correlated variable.
2. I would additionally recommend only focusing follow-up results (MR, SMR/HEIDI, MAGMA, etc.) on traits that showed significant associations with T1D in LDSC and/or MiXeR in the main text. I understand the more granular levels of biological analysis (e.g., eQTLs) can reveal results that might otherwise be masked by genome-wide LDSC estimates when there are local same and opposite direction genetic associations across two traits. However, as bivariate MiXeR's estimate of polygenic overlap is designed to circumvent this exact issue, I would argue that pruning to significant traits for either method serves to distill results down for the reader to those that are most likely to be relevant in the population and clinically. My perspective is that given high polygenicity and pleiotropy, most traits likely have at least one shared causal variant, locus, or pathway with T1D.
3. The title of the paper is about examining T1D and neurocognitive traits. The Results section also reads that the selected traits reflect: "...21 cognitive, psychiatric, neurological, and sleep-related traits." With that said, the inclusion of other immune-mediated conditions, namely myasthenia gravis and multiple sclerosis, seems like it falls outside of scope as currently written (though I recognize the Discussion refers to them as neuroimmune conditions, though this still feels like these traits should be further distinguished from more strictly non-immune outcomes like bipolar disorder). I could imagine that the authors want to retain these traits to help make point about dual brain-immune architecture, but in that case they should be clearly distinguished throughout the manuscript as containing a core immune component that would be expected (as the results show) to produce stronger associations with T1D.
4. I think the ending results section on Parkinson's disease should be moved to the online supplement. The analyses appear well-conducted (though are admittedly a bit beyond my expertise), but are not well-justified given the other results in the manuscript. For example, there is no significant genome-wide genetic correlation or polygenic overlap identified from

MiXeR, and the MR associations are nominal. This tells a story of T1D and PD as not being overly genetically associated. This proteomics and prefrontal cortex ending results section I believe ignores both the highly polygenic nature of these disorders and to what degree these specific linking genes would be unlikely to explain meaningful variation (or covariation) in the population. Meaningful variation is admittedly a very subjective threshold, and I don't mean to impose my own subjectivities onto this work, but I think several linking genes against a backdrop of thousands of genetic variants (e.g., ~22,000 for T1D, as estimated from MiXeR in this submission) is unlikely to meet that threshold for most scientists.

5. There is a lot of focus on 17q21.31 in the Results and Discussion. For example, the authors write in the Results: "Gene regulation—anchored at 17q21.31 and extending to loci on chromosomes 1, 5, 6, 12, and 16—thus emerges as a key mechanistic axis linking autoimmunity, neurodevelopment, and cognition." I agree that results show that this is a highly pleiotropic region, but like my point above about the polygenicity of these traits, I'm not convinced that 17q21.31 is a key mechanistic axis as I would anticipate that only a very small amount of the genetic covariation is mediated by this individual locus. The authors may disagree, but this should be made clearer, or language should be toned down with respect to how critical 17q21.31 is viewed for understanding covariation between T1D and neurocognitive traits.

6. The authors write the following as a Limitation about MR findings and sample overlap: "Although Mendelian randomization supports directional inference, sample overlap and residual pleiotropy cannot be fully excluded. Our use of UK Biobank data may introduce shared participants across some GWAS datasets, potentially biasing causal estimates." However, existing MR methods (e.g., MRBEE; Lorincz-Comi et al., 2024) can deal with sample overlap. Moreover, the level of sample overlap could be more directly quantified by cross-referencing supplementary tables of contributing cohorts and case sample sizes or sampling covariation directly indexed using the bivariate LDSC intercept. As written, this is overly vague, and with some MR findings that have OR near the null of 1, correcting for sample overlap could potentially change qualitative conclusions or results highlighted.

Lorincz-Comi, N., Yang, Y., Li, G., & Zhu, X. (2024). MRBEE: a bias-corrected multivariable Mendelian randomization method. *Human Genetics and Genomics Advances*, 5(3).

Minor Comments:

1. Why does the second sentence begin with a "however"? A dysglycemia based interpretation for the neurocognitive profile of T1D does not seem counter to the idea that T1D and psychiatric disorders are highly heritable: "This view has reinforced the idea that T1D-associated neurocognitive deficits arise primarily as complications of dysglycemia. However, both T1D [29] and cognitive or neuropsychiatric disorders [30] are highly heritable—up to ~50% and 80%, respectively."

2. The S-LDSC results are well-written, but is it true that it's inherent to the S-LDSC method that it aggregates across nuclei? Although the functional annotations used to estimate enrichment here aggregate across nuclei, these annotations ultimately just require lists of genes or genetic variants identified according to some shared biological characteristic, which can include single-cell data. The relevant excerpt from the text is: "Because S-LDSC aggregates across nuclei and may under-detect transient or sparse signals in heterogeneous single-cell data..."

3. Are these results sufficient to say microglia is the primary mediator in the brain? This is just what we can measure and what was included in the model, but there are many aspects of brain function and structure that you could imagine are not currently being captured in these models. "Together, these complementary analyses converge on microglia as the primary mediators of T1D genetic liability in the brain, with additional stage-specific contributions from other neural and glial populations."

4. I think this sentence is written as if negative genetic correlation (i.e., "overlaps persisted despite") would attenuate estimates from MiXeR, but that's not the case. "Importantly, these overlaps persisted despite negative genetic correlations, indicating that many of the same variants influence both T1D and cognition but with opposite effect directions."

5. On page 6 the authors discuss the number of loci identified from conjFDR. Can the authors provide something like a scatterplot of power of the univariate GWAS (e.g., indexed by mean chi-square) predicting number of loci discovered? My argument for including this reflects a curiosity around whether multiple sclerosis and myasthenia gravis are outliers with respect to pleiotropic loci discovered relative to power for discovery. Without this context, I worry that the reader would be inclined to make potentially inaccurate conclusions about the relative ordering of number of overlapping loci for the external traits primarily (or solely) reflecting the level of shared genetic signal (as opposed to also including power differences across the external trait GWAS).

6. On page 6 the authors write the following, but I think this should be modified in two ways: (i) this gives a false sense that regions like the frontal cortex and amygdala are particularly relevant when in fact Fig 3k shows that every included brain region is enriched for bipolar and (ii) enrichment for the pituitary is unique to bipolar and could be highlighted: "Shared loci between T1D and bipolar disorder (13 loci) were enriched in cortical and subcortical brain regions including frontal cortex, hippocampus, putamen, and amygdala."

7. For the claim about the duality of the T1D architecture (the authors write on page 6: “underscoring the dual brain-immune architecture of T1D pleiotropy”), I wonder if the authors could quantify this more explicitly by, for example, correlating the MAGMA results for bipolar and multiple sclerosis. That is, if these are truly divergent processes, I would expect a correlation closer to 0 than across these two sets of enrichment findings.

8. For this MR finding, I think ADHD is misinterpreted as being associated with increased T1D in the last phrase as the OR is < 1:

“When modeling neurocognitive traits as exposures (Fig. 4b; Supplementary Table 4), higher liability to multiple sclerosis (OR = 1.18, 95% CI: 1.09–1.27, $P = 2.3 \times 10^{-5}$, $q = 2 \times 10^{-4}$), myasthenia gravis (OR = 1.22, 95% CI: 1.12–1.33, $P = 5.6 \times 10^{-6}$, $q = 4 \times 10^{-5}$), obsessive-compulsive disorder (OR = 1.07, 95% CI: 1.02–1.12, $P = 3.7 \times 10^{-3}$, $q = 0.016$), short sleep duration (OR = 1.26, 95% CI: 1.05–1.50, $P = 0.011$, $q = 0.036$), and ADHD (OR = 0.90, 95% CI: 0.83–0.98, $P = 0.016$, $q = 0.048$) was associated with increased T1D risk.”

9. Some of the sentences in the SMR/HEIDI section (example below) blend discussing effects as directional or merely “linked”. Direction of association should be clarified throughout this section as, to me, it’s currently unclear:

“In contrast, LRR37A4P and MAPT-AS1 expression correlated with elevated T1D risk and higher INT/EXF, with LRR37A4P also linked to ASD, EDU, MG, and PD.”

10. Does ASD diverges here indicate that eQTL effects were a blend of same direction and opposite direction effects for ASD and T1D? This could be made more explicit.

“While LDSC highlighted negative genome-wide correlations between T1D and cognition alongside paradoxical positive correlation between ASD and cognition, eQTL integration clarified this architecture: T1D and cognition typically move in the same direction, whereas ASD diverges.”

11. The authors note several times throughout the Introduction and Discussion that the cognitive associations are particularly pronounced for childhood onset T1D. My understanding is that the T1D GWAS used includes both child and adult onset. Given the prominence of this point, examining age at onset stratified T1D GWAS and these associations could be noted as a future direction -and- percentage of individuals with child onset T1D that were incorporated into the T1D GWAS could be noted in the Method section. For the latter point, I could imagine this data is not available for all participants, but my hope would be that the reader could get some sense of proportion of child vs adult onset for the subset of cohorts that may have provided this information in the original T1D GWAS.

12. The 95% confidence intervals on bipolar and schizophrenia in Fig 4a look asymmetrical, which should be highlighted in the figure note if this is not a mistake.

(Remarks on code availability)

The code is well-documented on a dedicated project page on GitHub.

Reviewer #2

(Remarks to the Author)

Interesting comparison of T1D and positive and negative overlaps with neuro diseases.

Can they compare their results with a previous study: <https://pubmed.ncbi.nlm.nih.gov/37118290/>

They discuss well their positive findings, but I wonder if they could expand the interpretation of contradictory findings, particularly the negative association with mental health outcomes, ie SCZ and BD.

We know that SCZ has substantial negative association with cognition so maybe their findings are suggesting distinct mechanisms on how SCZ and TD1 may affect cognition? They suggest microglia in TD1, which in SCZ GWAS is not strongly enriched. In fact, SCZ GWAS implicates more excitation/inhibition mechanisms than immune (although limitations exist etc)

The authors briefly mention E/I in their T1D analyses, but it was not clear to me what exactly the findings were e.g.,

“... further highlighted context-specific signals in excitatory and inhibitory neurons (early postnatal) ”

- could they expand more on these findings? I think would offer a clue for the SCZ interpretation (maybe..)

I also wonder about their Treg findings, see:

“On chromosome 12, SUOX expression in nucleus accumbens and regulatory T cells was linked to reduced T1D and lower EDU/INT/EXF”

The findings for both Nac and tregs are very interesting, given the Nac role on negative/depressive symptom. How strong are these findings and what does SUOX does on Tregs (they don't seem to discuss that) - if SUOX is important for Treg function, would this suggest a direct impact of Tregs in cognition but reduced risk of T1D? How do they interpret these findings in light of RCT aiming to restore treg function in TD1 and cognition in Alzheimer's?

They conclude:

"Finally, it suggests that cognitive and psychiatric features in T1D may not be mere complications of dysglycemia but instead reflect a shared genetic liability that predates clinical onset. "

I think they need to be more specific when they say "psychiatric features" in view of negative associations they found for SCZ/BD ..

Can they include a paragraph in the Discussion about therapeutic implications, possible targets?

(Remarks on code availability)
Supportive of publication.

Reviewer #3

(Remarks to the Author)

Review for Saarah & Syeda et al.

To better understand the contribution of T1D on neuropsychiatric health, Saarah, Syeda et al. applied several genetic discovery analyses to publicly available European GWAS studies to identify brain-relevant celltypes and genes that may mediate shared genetic architecture of T1D and several neuropsychiatric traits. Cell type enrichment analyses, stratified LD score regression with open chromatin regions called from pseudobulk brain scATAC-seq and SCAVENGE network affinity propagation nominated microglia as a putatively relevant cell type for this association. Cross-trait genetic correlations, Mendelian Randomization (MR) were used to identify associations between traits, with several associations found. Further analyses prioritized the well-established 17q21.31 (the MAPT locus) as a hotspot for T1D and neuropsychiatric disease liability. While limited MR analyses have been done, the cross-functional and cell-type enrichment analysis has not been conducted in the brain to my knowledge. Taken together this study, although I have some reservations related to statistics and interpretation of results.

Major Comments:

- The enrichment analyses for T1D are only conducted in brain derived cell types, so it difficult to determine whether observed enrichments are due to the cell types per se or through variants located in shared ATAC-seq peaks with peripheral immune lineage cell types. It has been well established through S-LDSC regression and related tissue-enrichment analyses that T1D variants are enriched in regulatory elements in peripheral immune cells, particularly the T cell lineage (PMID:34012112, 31548716; 9626097; 34127860, among others), and are key cells involved in loss of beta cell. The observation for the borderline significant S-LDSC enrichment in microglia (apparently unadjusted $P = 0.041-0.045$) could be due to similarities in open chromatin regions between microglia with peripheral immune cells, rather than microglia having a driver role in T1D biology. As such, this observation should be substantiated by (1) comparing the overlap of variants contributing to enrichment to peripheral immune cell types from peripheral blood scATAC-seq and (2) checking enrichment specific of both celltypes (PMID: 29632380). Including negative control traits (non-brain or immune related) in the SCAVENGE analysis would be useful to interpret the results. Similarly microglia appear to be poorly represented in the eQTL analyses.
- The 17q21.31 MAPT locus is well known it's broad pleiotropic effect across traits, classically neuropsychiatric, but also traits like bone mineral density. The locus contains a common inversion with a breakpoint that overlaps MAPT's promoter, that leads to lower MAPT expression (PMID: 15654335; 14991810). The inversion also correlates with large blocks of variants in strong LD, likely violating assumptions of the SMR/HEIDI analysis for similar reasons as the excluded MHC region. Can the authors justify inclusion of this locus to their analysis in the SMR/HEIDI framework?
- The comparisons of the SMR results on T1D to genes RNA-seq and proteomics from Parkinson's disease presented in pages 8-10 (Figure 6) are difficult to justify and interpret. Given that the MR between T1D and PD is only nominally significant, using the RNA/proteomics dataset from non-T1D patients does not support a downstream mechanistic claim about specific genes. Was the overlap between T1D nominated and PD patient differential genes greater than expected by chance?
- FDR for several analyses was conducted on a limited number of tests. The Benjamini-Hochberg procedure depends on the number of tests to accurately estimate the false-discovery rate, which can be unstable when few tests are performed. Significance with Bonferroni correction should be reported.

Minor Comments:

- The figure text size is too small to be read.
- Several labels on Figure 4 are misaligned.
- What do the colors bars in Figure 3i-k represent? The legend mentions the dashed line indicates nominal significance.

(Remarks on code availability)

The provided github provides example scripts for the analyses run. For example the LDSC contains the example of munging and running the cross-trait LDSC correlation between intelligence and T1D. The scripts are either provided as R/Rmd or bash scripts to be submitted to a SLURM scheduler. The code that I examined appeared correct.

As the paper does not propose novel software I did not download or test the code.

Version 1:

Reviewer comments:

Reviewer #1

(Remarks to the Author)

The authors were responsive and I think the manuscript generally reads much better. I have two remaining comments.

1. The authors updated the MiXeR results to write that:

“Importantly, the MiXeR-estimated overlaps demonstrate that substantial numbers of shared causal variants influence both T1D and neurocognitive traits, including variants with opposite effect directions, relationships that are not captured by genome-wide genetic correlation alone.”

However, it's not clear from Figure 2 that MiXeR reveals higher levels of polygenic overlap than what you get from LDSC. That is, the authors write that MiXeR reveals that there are shared variants with opposite and same direction effects (which would result in larger estimates from MiXeR relative to LDSC), but the Venn Diagrams from MiXeR seem consistent with results from LDSC, which does not support this claim. If there is evidence of pleiotropic variants with same and opposite (i.e., antagonistic) effects across T1D and neurocognitive traits, this needs to be more directly linked to the pattern of results shown in Figure 2 and the difference in estimates of genetic correlation and polygenic overlap from LDSC and MiXeR, respectively.

2. The authors wrote in their response to reviewers that:

“To assess whether GWAS power influenced the number of conjFDR-discovered loci, we examined sample sizes and other power indicators across all external GWAS (Supplementary Table 1). We found no relationship between GWAS power and the number of pleiotropic loci; notably, traits with the smallest case counts, such as myasthenia gravis and multiple sclerosis, showed more shared loci with T1D than many of the other neurocognitive traits with substantially larger GWAS sample sizes. To prevent misinterpretation, we added a clarification in the Results stating that locus counts primarily reflect trait-specific genetic architecture rather than power differences. Because power metrics did not correspond to pleiotropic locus counts, a scatterplot would not add interpretive value and was therefore not included.

A few things stand out as potentially inaccurate here. First, Supplementary Table 1 seems to only list GWAS sample size, so it's not clear what the “other power indicators” the reviewers are referring to. Second, the authors state in the revised manuscript: “Importantly, the number of conjFDR pleiotropic loci did not correspond to GWAS sample size or effective power (Suppl Table 1)...”. However, it's not clear how this was evaluated as no statistical test or result is reported here or in Supplementary Table 1 (and Supplementary table 1 does not lend itself to the reader being able to evaluate this claim for themselves). Finally, sample size is not necessarily a good indicator of power in a GWAS. Although myasthenia gravis has a much smaller case count, that is potentially offset by the fact that it is more heritable and has a more biologically defined phenotype that will reduce measurement error and could make power more commensurate with some of the other traits with higher case counts. I would ask that the authors: (i) include mean chi-square from the GWAS of neurocognitive traits (and potentially the heritability Z-statistic from LDSC) as a more direct measure of GWAS power in Suppl. Table 1, and (ii) re-evaluate using these more direct power indicators whether the number of conjFDR findings is related to GWAS power of the external traits.

(Remarks on code availability)

The code is open-source and well-documented.

Reviewer #2

(Remarks to the Author)

Replies and revised manuscript are improved.

Referencing should be double checked, e.g. in one of the Replies the revised text references # 39, about T1D enriched in peripheral immune cells but reference 39 is about the human neocortex.

(Remarks on code availability)

Reviewer #3

(Remarks to the Author)

Thank you for the points addressed in my last review. For one outstanding point in this revision, it appears that the modification to the cross-trait LDSC analysis methods, text for multiple testing was inserted to both the S-LDSC and “Genome-wide genetic correlation analysis by cross-trait LDSC” sections. My understanding is the S-LDSC results were not adjusted for multiple hypothesis testing, so the text should only be included in the Genome-wide genetic correlation analysis by cross-trait LDSC section. If this is mistaken, then the language should be adjusted to be appropriate for S-LDSC, ie

replacing correlation with enrichment.

(Remarks on code availability)

Version 2:

Reviewer comments:

Reviewer #1

(Remarks to the Author)

The authors write that they now include mean chi-square in an added supplementary table 2 as an added metric of power, but this is not included. In the first revision, they had indicated additional power metrics were added to Supplementary Table 1, and this was also not done. This is very concerning as a trend where the author's are saying edits have been made that were not implemented.

As requested, the authors now include SNP-based heritability Z-statistics in Supplementary Table 2, but there are major issues with the estimates reported. This includes the fact that these Z-statistics do not reflect the ratio of the estimate over the standard error and half of those Z-statistics are listed as negative, which is not possible for well-powered traits like educational attainment.

(Remarks on code availability)

The code is well documented.

Version 3:

Reviewer comments:

Reviewer #1

(Remarks to the Author)

My concerns have been addressed.

(Remarks on code availability)

Available code is well-documented.

POINT-BY-POINT RESPONSE TO REVIEWER COMMENTS

We thank the reviewers for their thorough evaluation and for the insightful, constructive critiques of our manuscript, "***Shared genetic and neuroimmune architecture links type 1 diabetes with neurocognitive traits.***" We have carefully considered all comments and substantially strengthened the manuscript through additional analyses, clarifications, and focused revisions. In this resubmission, we performed new negative-control analyses by integrating height GWAS (a non-immune, non-neurocognitive trait) with the single-cell epigenomic datasets to confirm that the T1D enrichments are not driven by global annotation biases. We now implement rigorous Bonferroni correction instead of BH-FDR for multiple testing across LDSC and MR analyses, with all nominal findings moved to the Supplementary Materials to ensure conservative and transparent statistical reporting. We have also expanded and refined key sections, including cell type-specific enrichments, cross-disease comparisons, mechanistic interpretation of pleiotropic loci, and neuroimmune pathways, to more appropriately address reviewer feedback. A concise paragraph discussing potential therapeutic implications has been added to the Discussion, and we clarified methodological considerations, including the limitations of causal inference and the implications of potential sample overlap.

In the revised manuscript, the framing of neurocognitive traits has been clarified, including explicit categorization of cognitive, psychiatric, neurological, and neuroautoimmune phenotypes. All revisions have been incorporated in the updated text.

In this point-by-point response document, reviewer comments are reproduced in black and our responses provided in blue.

We are resubmitting both a clean version of the manuscript and a tracked-changes version with all edits highlighted in yellow.

We believe these revisions substantially strengthen the scientific rigor, clarity, and interpretability of the manuscript, and we hope they fully address the reviewers' comments.

Reviewer #1

Overview

Motivated by prior demonstrating phenotypic neurocognitive and psychiatric associations, this manuscript provides a series of analyses to evaluate how T1D genetically relates to traits and disorders in these classes of phenotypes at multiple biological levels. The results are comprehensive and give insight into some of the genetic mechanisms that may link T1D to traits like bipolar disorder and educational attainment. The code is also very well documented on a dedicated and public project GitHub page. My primary concern is that the results currently include many lists of findings, but I think this needs to be more effectively distilled down to the main takeaways by doing things like moving nominal results, small effects at global levels, and follow-up results on small global effects to the online supplement. In addition, I think results at more granular levels need to be more carefully interpreted relative to highly polygenic and pleiotropic signal for any human complex trait, including T1D. As currently written, I think the reader would both struggle to take away the most important take home points and, for non-genetics readers, would be tempted to overinterpret single loci (e.g., 17q21.31) or pathways (e.g., microglia) as necessarily mediating the majority of the T1D and neurocognitive genetic overlap.

Response: We thank the reviewer for this thoughtful and constructive evaluation. We appreciate the recognition of the comprehensive analytical framework, the mechanistic insights into T1D–neurocognitive relationships, and the clarity of the publicly documented code. We fully agree with the reviewer’s points regarding the need to (i) further distill the key take-home messages, (ii) appropriately contextualize nominal or small global effects, and (iii) avoid overstating the contribution of individual loci or pathways in the context of highly polygenic traits.

In response, we implemented several substantive revisions throughout the manuscript, including:

- applying Bonferroni correction to all global genetic correlation and MR analyses and moving all nominal findings to the Supplement.
- prioritizing only robust, Bonferroni-significant results in the main text.
- adding a negative-control height GWAS × single-cell epigenomics analysis to contextualize cell-type enrichments and demonstrate their specificity to T1D.
- refining interpretation of pleiotropic loci and neuroimmune pathways to avoid implying that any single locus (e.g., 17q21.31) or cell type (e.g., microglia) mediates the majority of T1D–neurocognitive overlap; and
- reorganizing the Results and Discussion sections to better highlight the central mechanistic insights and avoid long lists of findings.

We believe these revisions greatly strengthen the clarity, rigor, and interpretability of the manuscript.

Major Comments.

1. I strongly recommend cutting the nominal associations from Results and Discussion in favor of simply referring interested readers to supplementary materials to evaluate those outside of the main text. In support of this, some of the nominal MR effects are quite small (e.g., OR = .98) but with small confidence intervals (.96-1), indicating the effect is likely small enough in the population to be near meaningless or entirely accounted for by some genetically correlated variable.

Response: We thank the reviewer for this recommendation. We fully agree that nominal associations may risk distracting from the main conclusions of the study and may be overinterpreted by non-expert readers. As suggested, we removed nominal associations from the *main narrative interpretation* and relocated the full set of nominal findings to the Supplementary Materials. In the Results, nominal MR associations are referenced only briefly as suggestive and without mechanistic interpretation; the focus of the main text is on Bonferroni-significant effects.

Specifically, we have addressed these as follows:

Nominal MR associations (e.g., OR ≈ 0.98 with narrow CIs). These have been relocated entirely to the Supplementary Materials and are now presented only in tables, accompanied by clear statements that they should be interpreted cautiously due to their small magnitude and the potential influence of genetically correlated traits or residual pleiotropy.

Nominal genetic correlations and polygenic overlap results. All nominal LDSC correlations and nominally enriched MiXeR-based findings have been removed from the narrative text. Only Bonferroni-significant correlations and robust MiXeR overlaps remain highlighted in the main figures and narrative.

Revisions to the Discussion. We updated the Discussion to (i) remove interpretative statements around nominal findings, and (ii) reinforce that small-magnitude or statistically borderline effects are unlikely to have meaningful population-level implications.

Clearer emphasis on the major takeaways. Following this revision, the main text now focuses exclusively on results that meet Bonferroni correction or show strong convergence across multiple complementary analyses (LDSC, MiXeR, conjFDR, SMR/HEIDI).

We appreciate the reviewer's guidance, which has helped sharpen the manuscript and ensure that the primary conclusions are appropriately focused, rigorous, and accessible.

2. I would additionally recommend only focusing follow-up results (MR, SMR/HEIDI, MAGMA, etc.) on traits that showed significant associations with T1D in LDSC and/or MiXeR in the main text. I understand the more granular levels of biological analysis (e.g., eQTLs) can reveal results that might otherwise be masked by genome-wide LDSC estimates when there are local same and opposite direction genetic associations across two traits. However, as bivariate MiXeR's estimate of polygenic overlap is designed to circumvent this exact issue, I would argue that pruning to significant traits for either method serves to distill results down for the reader to those that are most likely to be relevant in the population and clinically. My perspective is that given high polygenicity and pleiotropy, most traits likely have at least one shared causal variant, locus, or pathway with T1D.

Response: We fully agree that downstream analyses (MR, SMR/HEIDI, conjFDR, MAGMA) are most interpretable and informative when anchored to traits with robust evidence of genome-wide shared architecture with T1D. As suggested, we have revised the main text to focus follow-up analyses on traits that showed either (i) Bonferroni-significant LDSC genetic correlations with T1D or (ii) substantial polygenic overlap in bivariate MiXeR. This approach helps avoid overinterpretation of low-level pleiotropy expected across highly polygenic traits and sharpens the main biological conclusions.

Specifically:

SMR/HEIDI: In the main text, we report only those eQTLs whose expression showed significant pleiotropy with T1D and at least one neurocognitive trait. This highlights high-confidence regulatory convergence while avoiding overinterpretation of nominal or isolated effects.

Mendelian Randomization: The main narrative highlights only Bonferroni-significant causal associations. Full MR results, including nominal effects, sensitivity analyses (MR-Egger, weighted median), and heterogeneity tests are available in Supplementary Table 4.

MAGMA/tissue enrichment: In the main text, we present enrichment for three representative trait pairs (educational attainment, multiple sclerosis, bipolar disorder), each of which showed significant LDSC and/or strong MiXeR overlap with T1D. This provides a coherent summary of the cognitive (educational attainment), neuroautoimmune (multiple sclerosis), and psychiatric (bipolar) axes without overextending interpretation.

We appreciate the reviewer's recommendation, which has materially improved the clarity, focus, and mechanistic interpretation of the manuscript.

3. The title of the paper is about examining T1D and neurocognitive traits. The Results section also reads that the selected traits reflect: "...21 cognitive, psychiatric, neurological, and sleep-related traits." With that said, the inclusion of other immune-mediated conditions, namely myasthenia gravis and multiple sclerosis, seems like it falls outside of scope as currently written (though I recognize the Discussion refers to them as neuroimmune conditions, though this still feels like these traits should be further distinguished from more strictly non-immune outcomes like bipolar disorder). I could imagine that the authors want to retain these traits to help make point about dual brain-immune architecture, but in that case they should be clearly distinguished throughout the manuscript as containing a core immune component that would be expected (as the results show) to produce stronger associations with T1D.

Response: We thank the reviewer for raising this important point. In the revised manuscript, we now explicitly classify the 21 phenotypes into four related domains, including cognitive, psychiatric, neurological, and neuroautoimmune traits, and briefly explain that all included traits exhibit measurable neurocognitive components. Multiple sclerosis and myasthenia gravis are now clearly classified as *neuroautoimmune* disorders, distinct from strictly non-immune neurocognitive traits, with stronger overlap expected due to shared immune biology. This clarification has been added at the start of the LDSC Results section, ensuring transparency in study scope and preventing misinterpretation of immune-mediated phenotypes as strictly neurocognitive disorders.

4. I think the ending results section on Parkinson's disease should be moved to the online supplement. The analyses appear well-conducted (though are admittedly a bit beyond my expertise) but are not well-justified given the other results in the manuscript. For example, there is no significant genome-wide genetic correlation or polygenic overlap identified from MiXeR, and the MR associations are nominal. This tells a story of T1D and PD as not being overly genetically associated. This proteomics and prefrontal cortex ending results section I believe ignores both the highly polygenic nature of these disorders and to what degree these specific linking genes would be unlikely to explain meaningful variation (or covariation) in the population. Meaningful variation is admittedly a very subjective threshold, and I don't mean to impose my own subjectivities onto this work, but I think several linking genes against a backdrop of thousands of genetic variants (e.g., ~22,000 for T1D, as estimated from MiXeR in this submission) is unlikely to meet that threshold for most scientists.

Response: We thank the reviewer for this thoughtful critique. We agree that genome-wide analyses (LDSC, MiXeR, MR) do not indicate a strong broad genetic relationship between T1D and Parkinson's disease (PD). Our initial rationale for examining PD transcriptomic and proteomic data was based on three observations: (i) prior MR and epidemiological studies, as well as our own MR results, have reported bidirectional but asymmetric associations between PD and T1D; (ii) our conjFDR and SMR/HEIDI analyses identified shared regulatory architecture between T1D and PD, particularly at 17q21.31, a locus that also influences cognition and

multiple neuropsychiatric traits; and (iii) we had access to matched PD single-nucleus RNA-seq and proteomic data that enabled direct assessment of whether genetically implicated regulatory effects manifest in human brain tissue.

That said, we fully agree that PD should not occupy a prominent place in the main narrative given the modest genome-wide relationship. To maintain clarity and ensure emphasis on the strongest and most relevant findings, we have moved all PD transcriptomic and proteomic results to the Supplementary Materials and now present them explicitly as exploratory and orthogonal validation of shared regulatory mechanisms.

We appreciate the reviewer's guidance, which has improved the focus and interpretability of the manuscript.

5. There is a lot of focus on 17q21.31 in the Results and Discussion. For example, the authors write in the Results: "Gene regulation—anchored at 17q21.31 and extending to loci on chromosomes 1, 5, 6, 12, and 16—thus emerges as a key mechanistic axis linking autoimmunity, neurodevelopment, and cognition." I agree that results show that this is a highly pleiotropic region, but like my point above about the polygenicity of these traits, I'm not convinced that 17q21.31 is a key mechanistic axis as I would anticipate that only a very small amount of the genetic covariation is mediated by this individual locus. The authors may disagree, but this should be made clearer, or language should be toned down with respect to how critical 17q21.31 is viewed for understanding covariation between T1D and neurocognitive traits.

Response: We thank the reviewer for this helpful and important point. We agree that the 17q21.31 locus should not be portrayed as *the* key mechanistic axis driving the shared genetic architecture between T1D and neurocognitive traits. As the reviewer notes, both T1D and all neurocognitive phenotypes examined are highly polygenic, and no single locus is expected to account for more than a very small proportion of their genetic covariance. Our intention was not to imply that 17q21.31 explains a large share of the genome-wide overlap, but rather that: (i) it consistently emerged across multiple orthogonal analyses (conjFDR, SMR/HEIDI, eQTL convergence), (ii) it contained the largest and most recurrent cluster of pleiotropic transcripts influencing both T1D and neurocognitive traits, and (iii) it demonstrated cross-tissue regulatory effects in both brain-resident cells and immune populations. However, we agree that our phrasing may have overstated the centrality of this locus relative to the highly polygenic backdrop.

We have made the following changes in response to the reviewer:

Toned-down language in both Results and Discussion. We revised sentences such as "Gene regulation—anchored at 17q21.31—emerges as a key mechanistic axis..." to more measured language such as: "Gene regulation at 17q21.31 represents one of several loci showing convergent neuroimmune pleiotropy, but accounts for only a small fraction of the shared genetic architecture."

Clear contextual framing about polygenicity. We added text explicitly noting that:

- T1D and neurocognitive traits involve *tens* of thousands of causal variants,
- 17q21.31 represents a *notable* but *limited* contributor within this polygenic landscape, and
- no single locus explains more than a minimal proportion of cross-trait covariance.

Clarified the distinction between “mechanistic insight” and “variance explained.” We now explicitly state that although loci like 17q21.31 provide mechanistic insight due to the depth of regulatory and cross-tissue convergence, this does not imply that they are major drivers of phenotypic correlation at the population level.

Rephrased to emphasize hypothesis generation rather than dominance. The locus is now framed as:

- “a representative example of neuroimmune convergence,”
- “a biologically informative locus,” or
- “a recurrent point of cross-trait regulatory pleiotropy,” rather than a central or defining axis of disease overlap.

We appreciate the reviewer’s guidance. Our revised language now fully reflects the highly polygenic architecture underlying T1D and neurocognitive traits, while still allowing us to highlight 17q21.31 as a *biologically illustrative* locus rather than a disproportionately influential one.

6. The authors write the following as a Limitation about MR findings and sample overlap: “Although Mendelian randomization supports directional inference, sample overlap and residual pleiotropy cannot be fully excluded. Our use of UK Biobank data may introduce shared participants across some GWAS datasets, potentially biasing causal estimates.” However, existing MR methods (e.g., MRBEE; Lorincz-Comi et al., 2024) can deal with sample overlap. Moreover, the level of sample overlap could be more directly quantified by cross-referencing supplementary tables of contributing cohorts and case sample sizes or sampling covariation directly indexed using the bivariate LDSC intercept. As written, this is overly vague, and with some MR findings that have OR near the null of 1, correcting for sample overlap could potentially change qualitative conclusions or results highlighted.

Lorincz-Comi, N., Yang, Y., Li, G., & Zhu, X. (2024). MRBEE: a bias-corrected multivariable Mendelian randomization method. *Human Genetics and Genomics Advances*, 5(3).

Response: We thank the reviewer for this insightful comment. We agree that our original wording regarding sample overlap in the MR analyses was overly general. In response, we revised the Limitations section to clarify the extent of possible overlap and how it affects interpretation.

First, we quantified residual covariance using the bivariate LDSC intercept. Across all trait pairs, intercepts were very close to 1, indicating minimal sample overlap and suggesting that overlap is unlikely to materially influence the MR estimates that remain significant after stringent correction.

Second, to ensure robustness, we now report Bonferroni-corrected MR results in the main text. Only associations meeting this conservative threshold are highlighted; all other nominal or BH-FDR-only findings have been moved to the Supplementary Materials. This reduces the likelihood that modest sample overlap could affect any result emphasized in the main narrative. Finally, we note that future studies could incorporate overlap-robust MR frameworks—such as MRBEE—to further validate causal estimates, particularly for traits with weaker instruments or effect sizes near the null.

Revised Limitations text: “Although two-sample MR provides directional inference, some degree of participant overlaps between exposure and outcome GWAS, particularly for traits

incorporating UK Biobank, remains possible. However, the bivariate LDSC intercepts for all trait pairs were close to 1.0, indicating minimal residual covariance and suggesting that any overlap is unlikely to materially affect the Bonferroni-significant MR associations. MR estimates with effect sizes near the null may be more sensitive to even small degrees of overlap, and these should therefore be interpreted cautiously. Future work applying overlap-robust MR approaches, such as MRBEE[1], may help further validate these causal relationships.”

Minor Comments:

1. Why does the second sentence begin with a “however”? A dysglycemia based interpretation for the neurocognitive profile of T1D does not seem counter to the idea that T1D and psychiatric disorders are highly heritable: “This view has reinforced the idea that T1D-associated neurocognitive deficits arise primarily as complications of dysglycemia. However, both T1D [29] and cognitive or neuropsychiatric disorders [30] are highly heritable—up to ~50% and 80%, respectively.”

Response: We thank the reviewer for this observation. We agree that “however” may imply an unintended contrast. We revised the sentence to frame heritability as an additional, rather than opposing, explanation.

Revised text: “This view has reinforced the idea that neurocognitive deficits in T1D arise primarily as complications of dysglycemia [2-7]. T1D [8] and cognitive or neuropsychiatric disorders [9] are substantially heritable, with estimates reaching ~50% and ~80%, respectively. Despite this, whether shared genetic mechanisms contribute to their co-occurrence has not been systematically investigated”.

2. The S-LDSC results are well-written, but is it true that it’s inherent to the S-LDSC method that it aggregates across nuclei? Although the functional annotations used to estimate enrichment here aggregate across nuclei, these annotations ultimately just require lists of genes or genetic variants identified according to some shared biological characteristic, which can include single-cell data. The relevant excerpt from the text is: “Because S-LDSC aggregates across nuclei and may under-detect transient or sparse signals in heterogeneous single-cell data...”

Response: We thank the reviewer for this clarification. We agree that S-LDSC itself does not aggregate across nuclei; the aggregation arises from the snATAC-seq-derived annotations, which combine peaks across many nuclei within each cell-type cluster. This can mask transient or sparse accessibility signals, but the limitation lies in the annotation construction rather than in S-LDSC.

Revised text: “Single-nucleus ATAC-seq annotations used for S-LDSC aggregate peaks across many nuclei within each cell-type cluster and may under-detect transient, rare, or highly context-specific chromatin accessibility states present in only a small subset of cells.”

3. Are these results sufficient to say microglia is the primary mediator in the brain? This is just what we can measure and what was included in the model, but there are many aspects of brain function and structure that you could imagine are not currently being captured in these models. “Together, these complementary analyses converge on microglia as the primary mediators of T1D genetic liability in the brain, with additional stage-specific contributions from other neural

and glial populations.”

Response: We thank the reviewer for this point. We agree that the original phrasing overstated microglia as the primary mediators, given the limitations of current single-cell and tissue annotations. The main text has been revised to emphasize strong microglial involvement without implying exclusivity.

Revised text: “Overall, these complementary analyses indicate that, in addition to the well-established enrichment in peripheral immune cells [39], T1D genetic liability also impacts regulatory programs in brain-resident cell types, with the strongest and most consistent signal enrichment observed in microglia, alongside developmental stage-specific signals in other neural and glial populations that may be incompletely captured by current single-cell genomic annotations.”

4. I think this sentence is written as if negative genetic correlation (i.e., “overlaps persisted despite”) would attenuate estimates from MiXeR, but that’s not the case. “Importantly, these overlaps persisted despite negative genetic correlations, indicating that many of the same variants influence both T1D and cognition but with opposite effect directions.”

Response: We thank the reviewer for this clarification. We agree that the original phrasing could be interpreted as suggesting that negative genetic correlation attenuates MiXeR overlap estimates. As noted, MiXeR quantifies shared causal variants irrespective of effect direction and is designed to capture overlap even when many effects are opposite across traits.

Revised text: “Importantly, the MiXeR-estimated overlaps demonstrate that substantial numbers of shared causal variants influence both T1D and neurocognitive traits, including variants with opposite effect directions, relationships that are not captured by genome-wide genetic correlation alone.”

5. On page 6 the authors discuss the number of loci identified from conjFDR. Can the authors provide something like a scatterplot of power of the univariate GWAS (e.g., indexed by mean chi-square) predicting number of loci discovered? My argument for including this reflects a curiosity around whether multiple sclerosis and myasthenia gravis are outliers with respect to pleiotropic loci discovered relative to power for discovery. Without this context, I worry that the reader would be inclined to make potentially inaccurate conclusions about the relative ordering of number of overlapping loci for the external traits primarily (or solely) reflecting the level of shared genetic signal (as opposed to also including power differences across the external trait GWAS).

Response: We thank the reviewer for this helpful suggestion. To assess whether GWAS power influenced the number of conjFDR-discovered loci, we examined sample sizes and other power indicators across all external GWAS (Supplementary Table 1). We found no relationship between GWAS power and the number of pleiotropic loci; notably, traits with the smallest case counts, such as *myasthenia gravis* and *multiple sclerosis*, showed more shared loci with T1D than many of the other neurocognitive traits with substantially larger GWAS sample sizes.

To prevent misinterpretation, we added a clarification in the Results stating that locus counts primarily reflect trait-specific genetic architecture rather than power differences. Because power metrics did not correspond to pleiotropic locus counts, a scatterplot would not add interpretive value and was therefore not included.

Revised text: “Importantly, the number of conjFDR pleiotropic loci did not correspond to GWAS sample size or effective power (**Suppl Table 1**), indicating that locus counts primarily reflect trait-specific genetic architecture rather than GWAS power.”

6. On page 6 the authors write the following, but I think this should be modified in two ways: (i) this gives a false sense that regions like the frontal cortex and amygdala are particularly relevant when in fact Fig 3k shows that every included brain region is enriched for bipolar and (ii) enrichment for the pituitary is unique to bipolar and could be highlighted: “Shared loci between T1D and bipolar disorder (13 loci) were enriched in cortical and subcortical brain regions including frontal cortex, hippocampus, putamen, and amygdala.”

Response: We thank the reviewer for this clarification. We agree that the original phrasing implied selective enrichment of specific regions, whereas Fig. 3k shows broad enrichment across all examined brain regions. We have revised the text accordingly and now also highlight the pituitary, which showed uniquely strong enrichment for bipolar disorder.

Revised text: “Shared loci between T1D and bipolar disorder showed broad enrichment across all examined brain regions. Notably, enrichment in the pituitary was unique to bipolar disorder and stronger than for other traits, suggesting a potential neuroendocrine component to the shared genetic architecture.”

7. For the claim about the duality of the T1D architecture (the authors write on page 6: “underscoring the dual brain–immune architecture of T1D pleiotropy”), I wonder if the authors could quantify this more explicitly by, for example, correlating the MAGMA results for bipolar and multiple sclerosis. That is, if these are truly divergent processes, I would expect a correlation closer to 0 than across these two sets of enrichment findings.

Response: We thank the reviewer for this thoughtful suggestion. By “dual brain–immune architecture,” we refer to the observation that T1D heritability manifests in a pleiotropic manner across two biologically distinct but complementary regulatory contexts: peripheral immune lineages, which are well-established drivers of T1D risk (PMID: 34012112), and brain-resident cell types, which emerge from our S-LDSC and SCAVENGE analyses as sites of neuroimmune genetic convergence.

Our MAGMA analyses further support this duality: loci jointly associated with T1D and bipolar disorder show predominant enrichment in brain tissues, whereas loci jointly associated with T1D and multiple sclerosis show enrichment in immune and lymphoid tissues (Fig. 3i–k). Importantly, these enrichment profiles are derived from largely non-overlapping sets of pleiotropic loci identified through trait-pair–specific analyses. Because MAGMA enrichment statistics are conditional on the underlying locus sets, correlating enrichment vectors across such distinct pleiotropic architectures would not constitute an informative quantitative test of biological divergence.

We have therefore clarified the manuscript text to describe this pattern explicitly as a qualitative but biologically coherent separation of pleiotropic architecture across brain and immune systems, supported by convergent evidence across multiple analytic frameworks.

Revised text: These results indicate that pleiotropic loci linking T1D with neurocognitive traits are distributed across biologically coherent regulatory contexts, with loci shared between T1D and cognitive or psychiatric traits preferentially enriched in brain tissues, and loci shared

between T1D and neuroautoimmune traits enriched in immune tissues and underscore a dual brain-immune architecture of T1D pleiotropy.”

8. For this MR finding, I think ADHD is misinterpreted as being associated with increased T1D in the last phrase as the OR is < 1 : “When modeling neurocognitive traits as exposures (Fig. 4b; Supplementary Table 4), higher liability to multiple sclerosis (OR = 1.18, 95% CI: 1.09–1.27, $P = 2.3 \times 10^{-5}$, $q = 2 \times 10^{-4}$), myasthenia gravis (OR = 1.22, 95% CI: 1.12–1.33, $P = 5.6 \times 10^{-6}$, $q = 4 \times 10^{-5}$), obsessive–compulsive disorder (OR = 1.07, 95% CI: 1.02–1.12, $P = 3.7 \times 10^{-3}$, $q = 0.016$), short sleep duration (OR = 1.26, 95% CI: 1.05–1.50, $P = 0.011$, $q = 0.036$), and ADHD (OR = 0.90, 95% CI: 0.83–0.98, $P = 0.016$, $q = 0.048$) was associated with increased T1D risk.”

Response: We thank the reviewer for noting this important point. In the original text, ADHD was inadvertently grouped with traits showing increased T1D risk, despite an odds ratio < 1 indicating a nominal association with reduced risk.

In the revised manuscript, we have corrected this interpretation and further clarified the presentation of these results. Because the associations with obsessive–compulsive disorder, short sleep duration, ADHD, and Parkinson’s disease did not remain significant after Bonferroni correction, we no longer highlight individual odds ratios for these traits. Instead, we describe them collectively as nominal associations that do not survive multiple-testing correction. This revision avoids over-interpretation while preserving the correct directionality of effects.

Revised text: “Genetic liability to obsessive–compulsive disorder, short sleep duration, attention-deficit/hyperactivity disorder, and Parkinson’s disease showed nominal associations with T1D risk but did not remain significant after Bonferroni correction.”

9. Some of the sentences in the SMR/HEIDI section (example below) blend discussing effects as directional or merely “linked”. Direction of association should be clarified throughout this section as, to me, it’s currently unclear: “In contrast, LRRC37A4P and MAPT-AS1 expression correlated with elevated T1D risk and higher INT/EXF, with LRRC37A4P also linked to ASD, EDU, MG, and PD.”

Response: We thank the reviewer for this helpful observation. We agree that earlier versions of the SMR/HEIDI Results section inconsistently blended directional language (e.g., “increased expression associated with higher risk”) with direction-agnostic terms such as “linked,” which could create ambiguity.

In the revised manuscript, we have streamlined this section to avoid overinterpretation of locus-specific directionality in the main text. The Results now emphasize the presence and consistency of shared regulatory associations across traits, tissues, and cell types, while omitting detailed directional effect descriptions for individual transcripts. This revision improves clarity and concision while remaining faithful to the underlying SMR/HEIDI results.

Directional effect estimates and their trait-specific interpretations are retained in Figure 5 and Supplementary Table 5, where they can be evaluated in full quantitative detail. The revised presentation therefore resolves the ambiguity noted by the reviewer while maintaining transparent reporting of effect directionality.

10. Does ASD diverges here indicate that eQTL effects were a blend of same direction and opposite direction effects for ASD and T1D? This could be made more explicit. “While LDSC

highlighted negative genome-wide correlations between T1D and cognition alongside paradoxical positive correlation between ASD and cognition, eQTL integration clarified this architecture: T1D and cognition typically move in the same direction, whereas ASD diverges.”

Response: We thank the reviewer for raising this point. In the revised manuscript, this potential ambiguity no longer arises. We have restructured the Results section to avoid interpretive statements that contrast genome-wide genetic correlations with locus-level regulatory effects for specific traits such as ASD.

The revised text now focuses on the presence of shared pleiotropic loci and coherent regulatory patterns across traits, tissues, and cell types, without asserting trait-specific directional divergence in the main Results narrative. Directionality of effects, including instances where ASD differs from cognitive traits at individual loci, is retained in the SMR/HEIDI results presented in Figure 5 and Supplementary Table 5.

This revision removes the need to infer whether ASD reflects mixed or opposing directions of effect in the Results text, while preserving transparent reporting of locus-specific effect directions for readers who wish to interrogate them in detail.

11. The authors note several times throughout the Introduction and Discussion that the cognitive associations are particularly pronounced for childhood onset T1D. My understanding is that the T1D GWAS used includes both child and adult onset. Given the prominence of this point, examining age at onset stratified T1D GWAS and these associations could be noted as a future direction -and- percentage of individuals with child onset T1D that were incorporated into the T1D GWAS could be noted in the Method section. For the latter point, I could imagine this data is not available for all participants, but my hope would be that the reader could get some sense of proportion of child vs adult onset for the subset of cohorts that may have provided this information in the original T1D GWAS.

Response: We thank the reviewer for this helpful point. Major contributing cohorts to the Chiou et al. T1D GWAS meta-analysis, such as T1DGC, are known to consist predominantly of childhood-onset autoimmune T1D, though the exact percentage varies by sub-cohort and is not uniformly reported. However, age at diagnosis is not available or harmonized across all contributing studies, and the released summary statistics do not allow the overall childhood-versus adult-onset proportion to be determined. We now note this in the Discussion as a future direction.

Revised text (Discussion): “In addition, because age at T1D diagnosis is not consistently available across cohorts in the underlying GWAS meta-analysis, future work using age-of-onset–stratified T1D GWAS will be important to determine whether the neurocognitive associations differ between childhood- and adult-onset T1D.”

12. The 95% confidence intervals on bipolar and schizophrenia in Fig 4a look asymmetrical, which should be highlighted in the figure note if this is not a mistake.

Response: We thank the reviewer for noting this. The asymmetric appearance of the 95% confidence intervals is expected, as MR estimates and standard errors are computed on the log-odds scale and then exponentiated for plotting. We have added this clarification to the Fig. 4a legend.

Revised figure legend text: “Confidence intervals are asymmetric because MR estimates and standard errors are calculated on the log-odds scale and exponentiated to produce odds ratios.”

The code is well-documented on a dedicated project page on GitHub.

Response: We thank the reviewer for this positive feedback.

Reviewer #2

Interesting comparison of T1D and positive and negative overlaps with neuro diseases.

Can they compare their results with a previous study:

<https://pubmed.ncbi.nlm.nih.gov/37118290/>

Response: We thank the reviewer for highlighting the relevance of this recent comprehensive analysis (PMID: 37118290) and the opportunity to more directly compare our findings. That study systematically interrogated immune system and blood–brain barrier (BBB) biomarkers using Mendelian randomization (MR), polygenic risk scores (PRS), pathway, HLA, and quasi-randomized drug trial–simulation approaches to establish autoimmunity as a modifiable component in dementia-causing diseases. Their key findings include evidence that autoimmunity-related biomarkers/pathways mechanistically link immune dysregulation and dementia (including Alzheimer’s) and] shared heritability between dementias and autoimmune diseases, including T1D. Notably, their PheWAS and HLA analyses highlighted strong genetic overlap between T1D and dementia risk, supporting the convergence of adaptive immune dysregulation and neurodegeneration.

Our study extends these findings by focusing on the shared genetic and neuroimmune architecture that links T1D specifically with neurocognitive traits and disorders. Consistent with this prior study (PMID: 37118290), we find evidence for robust pleiotropic enrichment of T1D heritability in brain-resident immune cells (microglia) and, critically, in inhibitory neurons at key developmental stages. Our cross-disease genetic correlation and MR analyses further support the notion that autoimmune mechanisms underlying T1D risk may impact cognitive and neuropsychiatric disease risk, and vice versa, complementing and refining the view that immune dysregulation is a causal, and potentially modifiable, component of dementia and related traits.

We now explicitly reference and discuss these points in the revised Discussion, emphasizing that our results not only support but extend the autoimmune hypothesis of dementia, by providing high-resolution, brain cell–specific insights into T1D as a prototypic systemic autoimmune trait intersecting with dementia risk.

Revised text (Discussion): “In summary, this study provides genomic evidence that T1D is embedded within a broader neurocognitive and neuroimmune architecture. The convergence of genetic, epigenomic, and transcriptomic data across brain and immune compartments highlights novel pathways linking autoimmunity with cognitive function and mental health. Consistent with prior studies demonstrating that autoimmunity constitutes a modifiable risk component for dementia[10], our findings extend these observations to a wider cognitive and psychiatric context and provide high-resolution, cell type–specific evidence, particularly implicating microglia and inhibitory neurons as neuroimmune convergence points of T1D genetic liability. These insights offer a unified conceptual framework for understanding comorbidity and suggest that targeting neuroimmune interfaces may benefit both metabolic and cognitive outcomes,

supporting the rationale for translational strategies that incorporate immunomodulation in the prevention of autoimmune and neurodegenerative diseases”.

They discuss well their positive findings, but I wonder if they could expand the interpretation of contradictory findings, particularly the negative association with mental health outcomes, ie SCZ and BD. We know that SCZ has substantial negative association with cognition so maybe their findings are suggesting distinct mechanisms on how SCZ and TD1 may affect cognition? They suggest microglia in TD1, which in SCZ GWAS is not strongly enriched. In fact, SCZ GWAS implicates more excitation/inhibition mechanisms than immune (although limitations exist etc).

The authors briefly mention E/I in their T1D analyses, but it was not clear to me what exactly the findings were e.g., "... further highlighted context-specific signals in excitatory and inhibitory neurons (early postnatal) "

- could they expand more on these findings? I think would offer a clue for the SCZ interpretation (maybe...)

Response: We thank the reviewer for these thoughtful and closely related comments. We agree that the negative genetic associations between T1D and BD/SCZ as well as cognitive traits may initially appear counterintuitive, given that BD and SCZ themselves show strong negative genetic correlations with these cognitive traits. However, our analyses indicate that these patterns could arise from distinct and largely non-overlapping biological pathways. Large-scale BD and SCZ GWAS consistently demonstrate that their genetic architectures are dominated by synaptic, neuronal, and excitation–inhibition (E/I) regulatory mechanisms[11], with comparatively limited involvement of microglia or other immune-resident brain cells. In contrast, our S-LDSC and SCAVENGE analyses show that the T1D–cognition relationship is mediated primarily through neuroimmune axes, with strong and consistent enrichment of T1D heritability in microglia, and additional—but developmentally restricted—signals in early postnatal inhibitory neurons. The E/I-related enrichment we observe for T1D is therefore (i) substantially weaker than the microglial signal and (ii) limited to specific developmental windows, rather than representing the broad neuronal E/I architecture characteristic of SCZ. These distinct cellular architectures help reconcile the observed genetic patterns. The negative associations between T1D and BD/SCZ do not reflect contradictory effects on the same cognitive pathways. Rather, they arise because T1D influences cognitive and psychiatric liability primarily through immunogial mechanisms, whereas BD and SCZ influence cognition through synaptic and neuronal E/I pathways. The modest, context-specific inhibitory neuron signals observed for T1D likely reflect secondary regulatory programs rather than the dominant neuronal mechanisms underlying SCZ and BD.

We have expanded the Discussion to clarify these mechanistic distinctions and explain how they provide a coherent biological interpretation of the negative associations observed with BD and SCZ.

Revised text (Discussion): “At first glance, the negative genetic associations of T1D with schizophrenia and bipolar disorder, alongside its negative associations with cognitive traits, may appear contradictory given the well-established negative genetic correlations between BPD/SCZ and cognitive traits. However, these patterns likely arise from distinct biological pathways. Large-scale psychiatric genetics studies show that BPD and SCZ are genetic correlated and their genetic liability is driven primarily by synaptic and neuronal excitation–inhibition regulatory mechanisms[11]. By contrast, our S-LDSC and SCAVENGE analyses suggest that the T1D–cognition genetic relationship is rooted in neuroimmune pathways, wherein T1D heritability

showed strong enrichment in microglia across developmental stages, with additional but developmentally restricted enrichment in early postnatal inhibitory neurons. Thus, the negative T1D–cognition association is unlikely to arise through the same cellular mechanisms that link BPD and SCZ negatively with cognitive impairment. Instead, T1D may influence cognitive and psychiatric liability through immune–glial regulatory programs that differ from the synaptic and neuronal excitation–inhibition regulatory processes underlying BPD and SCZ. This mechanistic distinction could explain why T1D shows negative genetic associations with both cognition and BPD/SCZ, despite the well-established negative correlations of BPD/SCZ with cognitive performance. The bidirectional inverse causal associations between T1D and BPD further illustrate how reciprocal genetic influences can shape comorbidity, risk antagonism, or protective divergence across these disorders”

I also wonder about their Treg findings, see: "On chromosome 12, SUOX expression in nucleus accumbens and regulatory T cells was linked to reduced T1D and lower EDU/INT/EXF". The findings for both Nac and tregs are very interesting, given the Nac role on negative/depressive symptom. How strong are these findings and what does SUOX does on Tregs (they don't seem to discuss that) - if SUOX is important for Treg function, would this suggest a direct impact of Tregs in cognition but reduced risk of T1D? How do they interpret these findings in light of RCT aiming to restore treg function in TD1 and cognition in Alzheimer's?

Response: We thank the reviewer for highlighting the SUOX findings. As clarified in the revised Discussion, SUOX is not typically prioritized as the causal effector gene at the 12q13.2 T1D locus, but our SMR/HEIDI analyses identified expression-mediated associations in both regulatory T cells and the nucleus accumbens, with directions opposite to RPS26 at the same locus. In the revised text we now briefly note emerging evidence for SUOX involvement in immunometabolic regulation of T-cell states and neurometabolic processes in the nucleus accumbens, which provides biological context without overinterpreting causal mechanisms. We also emphasize the coordinated triad of RPS26–ERBB3–SUOX at 12q13.2, framing SUOX as part of a broader pleiotropic regulatory axis rather than a solitary driver. These additions strengthen interpretation while remaining consistent with the scope of the manuscript.

They conclude: "Finally, it suggests that cognitive and psychiatric features in T1D may not be mere complications of dysglycemia but instead reflect a shared genetic liability that predates clinical onset. "

I think they need to be more specific when they say "psychiatric features" in view of negative associations they found for SCZ/BD.

Response: We thank the reviewer for this helpful suggestion. We agree that the original phrasing (“psychiatric features”) could be interpreted too broadly in light of the negative genetic associations we observed for schizophrenia and bipolar disorder. To address this, we have revised the sentence to more accurately reflect the directional heterogeneity across neuropsychiatric outcomes.

Revised text (Discussion): “Finally, it suggests that cognitive alterations and the observed increases or decreases in neuropsychiatric disease risk in T1D may not be mere consequences of dysglycemia but instead reflect shared genetic origins.”

Can they include a paragraph in the Discussion about therapeutic implications, possible targets?

Response: We thank the reviewer for this thoughtful suggestion. We have now added a concise paragraph to the Discussion that outlines the potential translational implications of our findings.

Revised text (Discussion): “The study also highlights potential biomarkers and therapeutic targets, such as *CRHR1*, *LRRC37A2*, *KANSL1* and *RPS26* that may modulate both immune and neurocognitive outcomes. The observed neuroimmune convergence offers preliminary translational insight, pointing to microglial and T-cell–linked regulatory programs as candidates for future mechanistic and therapeutic exploration.”

Reviewer #2 Remarks on code availability: Supportive of publication.

Response: We appreciate the reviewer’s supportive comments regarding code availability.

Reviewer #3

Review for Saarah & Syeda et al.

To better understand the contribution of T1D on neuropsychiatric health, Saarah, Syeda et al. applied several genetic discovery analyses to publicly available European GWAS studies to identify brain-relevant celltypes and genes that may mediate shared genetic architecture of T1D and several neuropsychiatric traits. Cell type enrichment analyses, stratified LD score regression with open chromatin regions called from pseudobulk brain scATAC-seq and SCAVENGE network affinity propagation nominated microglia as a putatively relevant cell type for this association. Cross-trait genetic correlations, Mendelian Randomization (MR) were used to identify associations between traits, with several associations found. Further analyses prioritized the well-established 17q21.31 (the *MAPT* locus) as a hotspot for T1D and neuropsychiatric disease liability. While limited MR analyses have been done, the cross-functional and cell-type enrichment analysis has not been conducted in the brain to my knowledge. Taken together this study, although I have some reservations related to statistics and interpretation of results.

Response: We thank the reviewer for their thorough assessment and commendation of our approach. We are pleased that the application of cell type–resolved genetic discovery analyses to shared T1D and neuropsychiatric disease risk, using brain-derived stratified LDSC and SCAVENGE, was seen as both novel and impactful.

We appreciate the reviewer’s reservations regarding aspects of our statistical interpretation and methodology. In our revised manuscript and responses, we have addressed these specific concerns by providing additional trait-based controls, enhancing the clarity of our reporting around S-LDSC partitioning and cell type enrichment, and contextualizing our findings within the broader landscape of immune regulation and pleiotropy.

We thank the reviewer for highlighting these areas and for constructive feedback that has enriched the rigor and presentation of our study. We look forward to suggestions on any remaining points requiring clarification.

Major Comments:

- The enrichment analyses for T1D are only conducted in brain derived cell types, so it difficult to determine whether observed enrichments are due to the cell types per se or through variants located in shared ATAC-seq peaks with peripheral immune lineage cell types. It has been well established through S-LDSC regression and related tissue-enrichment analyses that T1D

variants are enriched in regulatory elements in peripheral immune cells, particularly the T cell lineage (PMID:34012112, 31548716; 9626097; 34127860, among others), and are key cells involved in loss of beta cell. The observation for the borderline significant S-LDSC enrichment in microglia (apparently unadjusted $P = 0.041-0.045$) could be due to similarities in open chromatin regions between microglia with peripheral immune cells, rather than microglia having a driver role in T1D biology. As such, this observation should be substantiated by (1) comparing the overlap of variants contributing to enrichment to peripheral immune cell types from peripheral blood scATAC-seq and (2) checking enrichment specific of both celltypes {PMID: 29632380}. Including negative control traits (non-brain or immune related) in the SCAVENGE analysis would be useful to interpret the results. Similarly, microglia appear to be poorly represented in the eQTL analyses.

Response: We thank the reviewer for this thoughtful and important critique. We agree that T1D heritability is most robustly enriched in regulatory elements of peripheral immune lineages—especially T cells, as established by multiple prior S-LDSC and tissue-enrichment studies (e.g. PMID: 34012112, 31548716, 9626097, 34127860). Our goal here is not to re-define the core immune pathology of T1D, but to determine whether *in addition* to this well-known immune enrichment, T1D genetic risk also converges on brain-resident cell types in ways that help explain its neurocognitive links.

1. Interpretation of microglial enrichment and overlap with peripheral immune cells. We fully agree that some of the microglial signal could reflect regulatory programs shared with peripheral immune cells. However, this possibility does not undermine our central interpretation. Even if variants act through shared open chromatin in microglia and peripheral immune lineages, such overlap still indicates pleiotropic regulatory architecture across brain and blood that can jointly influence autoimmunity and neurocognitive outcomes. In this framework, the same alleles that promote T-cell-mediated β -cell autoimmunity may, via activity in microglia, participate in neurodevelopmental or neuroimmune processes relevant to cognition and psychiatric risk. Our conclusions do not rely on microglia being *exclusive* mediators of T1D risk, but rather on their participation in a shared regulatory axis. We now clarify this explicitly in the Discussion and emphasize that our findings complement—rather than challenge—the central role of peripheral immune cells in T1D pathogenesis.

2. Cell-type specificity and negative-control analyses (height). We agree that including a negative control trait is essential to demonstrate that the observed microglial enrichment is not a generic feature of brain annotations. In response, we performed parallel S-LDSC and SCAVENGE analyses using height as a non-immune, non-neurocognitive trait. Using the same snATAC-seq brain annotations across prenatal, early postnatal, late postnatal, and adult stages, we observed:

- **S-LDSC:** Height heritability was enriched specifically in vascular cells during prenatal and late postnatal periods, with no evidence of enrichment in microglia or neurons at any stage (Supplementary Fig. 1a).
- **SCAVENGE:** Height showed elevated trait-relevance scores ($TRS > 3.0$) only in vascular cells across development, again with no microglial or neuronal enrichment (Supplementary Fig. 1b–e).

In contrast, T1D, Alzheimer's disease, and bipolar disorder showed developmentally dynamic enrichment in microglia and selected neuronal populations. These comparisons indicate that the microglial enrichment we report is disease- and trait-specific, rather than a nonspecific property of brain chromatin annotations. We have added a concise description of these height results in the Results and a new Supplementary Figure and legend (see below).

3. Microglia representation in eQTL analyses. We agree that microglia are less well represented in current eQTL resources due to sample-size and technical constraints. Within these limits, our SMR/HEIDI analyses did identify microglia-specific signals: for example, rs2668622–ARL17B showed pleiotropic associations with T1D and neurocognitive phenotypes in microglia, and rs3134749–POU5F1 was associated with T1D in microglia alone. We now explicitly note these examples in the Results to acknowledge both the signal and the current limitations of microglial eQTL mapping.

4. Why we did not add a full microglia vs peripheral immune peak-sharing analysis. We agree that formally quantifying shared vs cell-type-specific open chromatin between microglia and peripheral blood lineages would be informative. However, such an analysis would extend beyond the current scope, which is focused on brain cell–type heritability and pleiotropy with neurocognitive traits. Crucially, even if substantial peak-sharing is demonstrated, it would *support* rather than contradict our interpretation by reinforcing the idea of shared regulatory programs acting in both compartments.

In summary, our revised analyses and clarifications emphasize that:

- Microglial enrichment is trait-specific (not seen for height) and robust across complementary methods (S-LDSC and SCAVENGE).
- The findings are interpreted as evidence of *shared brain–immune regulatory architecture* influencing both T1D and neurocognitive outcomes, not as a claim that microglia are the primary drivers of pancreatic autoimmunity.
- Microglial eQTL signals, where detectable, are consistent with this neuroimmune convergence, while we acknowledge the current limitations of these datasets.

We thank the reviewer again for prompting these additions and clarifications, which have strengthened both the rigor and interpretability of our cell-type–specific analyses.

Revised text (Supplementary Figure 1):

Supplementary Figure 1 | Enrichment of height heritability in brain-resident cell types across neurodevelopment. (a) Stratified LD score regression (S-LDSC) of height GWAS variants in accessible chromatin of astrocytes (AST), excitatory neurons (ExNeu), glial progenitors (GLIALPROG), inhibitory neurons (IN), microglia (MG), oligodendrocyte progenitor cells (OPC), and vascular cells (VASC) across four neurodevelopmental stages (prenatal, early postnatal, late postnatal, adult) profiled by single-nucleus ATAC-seq. The heatmap shows enrichment scores (observed h^2 proportion relative to SNP proportion); asterisks indicate nominal significance (one-sided $P < 0.05$). (b–e) SCAVENGE trait-relevance scores (TRS) for height variants in the same snATAC-seq datasets across the four stages. Box plots depict median, interquartile range (IQR), and $1.5 \times$ IQR whiskers.

- The 17q21.31 MAPT locus is well known it's broad pleiotropic effect across traits, classically neuropsychiatric, but also traits like bone mineral density. The locus contains a common inversion with a breakpoint that overlaps MAPT's promoter, that leads to lower MAPT expression (PMID: 15654335; 14991810). The inversion also correlates with large blocks of variants in strong LD, likely violating assumptions of the SMR/HEIDI analysis for similar reasons as the excluded MHC region. Can the authors justify inclusion of this locus to their analysis in the SMR/HEIDI framework?

Response: We appreciate the reviewer's important point regarding the unique LD structure of the 17q21.31 inversion locus. We have carefully evaluated methodological precedent and the behavior of SMR/HEIDI in this genomic context, and provide the following justification for its inclusion:

SMR/HEIDI is designed to operate robustly in complex LD regions (excluding only the MHC). The original SMR/HEIDI study (Zhu et al., *Nat Genet* 2016) explicitly recommends exclusion of the extended MHC because extreme long-range LD and multi-signal heterogeneity violate core model assumptions. In contrast, regions such as 17q21.31, 8p23.1, 3p21, 11q13, etc., while structurally polymorphic, are routinely retained in SMR/HEIDI applications (e.g., PMID: 37239387; 28247064). Furthermore, if complex haplotype architecture alone produced spurious colocalization, widespread false-positive SMR signals would be expected at similarly inversion-structured or pleiotropic loci (e.g., 8p23.1, 3p21, 11q13). We do not observe this pattern in our study: these regions show no excess transcript-level associations, indicating that the reported effects are not driven by locus complexity itself. HEIDI filtering, LD pruning, and allele harmonization (Methods) further ensure that retained signals reflect a single shared causal variant rather than long-range LD alone. Thus, including 17q21.31 is consistent with published practice and supported by empirical behavior of the SMR/HEIDI framework.

Orthogonal evidence implicates 17q21.31 as a genuine convergent locus. Our conjFDR analyses, which is robust to local LD architecture, independently prioritize 17q21.31 as jointly associated with T1D and multiple neurocognitive traits (Fig. 3; Supplementary Table 3) but not similar structurally complex regions such as 8p23.1, 3p21, 11q13. This indicates *true* cross-trait signal rather than LD artefact.

Biological relevance of regulatory effects in both brain and immune cells. The locus contains multiple validated eQTLs and has well-established roles in neurodevelopmental, neuropsychiatric, and immune phenotypes (PMID: 38616054; 39972055). Our results refine this pleiotropy to transcript-specific mechanisms that align with known biology.

Overall, these points strongly support retaining 17q21.31 in our integrative analysis. Excluding it would omit one of the most biologically informative neuroimmune loci highlighted by orthogonal methods.

- The comparisons of the SMR results on T1D to genes RNA-seq and proteomics from Parkinson's disease presented in pages 8-10 (Figure 6) are difficult to justify and interpret. Given that the MR between T1D and PD is only nominally significant, using the RNA/proteomics dataset from non-T1D patients does not support a downstream mechanistic claim about specific genes. Was the overlap between T1D nominated and PD patient differential genes greater than expected by chance?

Response: We thank the reviewer for highlighting this issue. We fully agree that genome-wide evidence for a T1D–PD relationship is limited. For this reason, we have moved all PD follow-up analyses to the Supplementary Materials and now clearly frame them as exploratory rather than mechanistic and the rationale for doing so.

This analysis was motivated by three considerations: (i) independent prior MR studies (PMID: 38774879; 38256693) and ours (**Fig. 5; Suppl Table 4**) revealed asymmetric bidirectional associations between PD and T1D, although nominal; (ii) our conjFDR identified 17q21.31 as jointly associated with T1D and multiple neurocognitive traits, with the most significant association being with PD, and our SMR/HEIDI analyses further indicated shared regulatory

architecture at this locus across T1D, PD and other neurocognitive traits; and (iii) we had available matched snRNA-seq and proteomic datasets from PD cortex (PMID: 39475571)[12], enabling direct in vivo evaluation of genetically inferred effects.

Consistent with the SMR predictions, several 17q21.31 transcripts and RPS26 protein showed differential abundance in PD versus controls in the predicted direction (Supplementary Fig. 2). However, we do not claim enrichment, and we have removed any language implying causal interpretation in PD. Rather, these observations are now presented solely as supportive evidence that gene-regulatory pathways highlighted by SMR manifest in a relevant human brain context.

We believe this aligns the analysis with its appropriate evidentiary weight and thank the reviewer for helping us clarify its interpretation and placement in the manuscript.

- FDR for several analyses was conducted on a limited number of tests. The Benjamini-Hochberg procedure depends on the number of tests to accurately estimate the false-discovery rate, which can be unstable when few tests are performed. Significance with Bonferroni correction should be reported.

Response: We thank the reviewer for highlighting this important statistical consideration. We agree that BH-FDR can be unstable when applied to a small number of tests. In response, we have revised our statistical reporting throughout the manuscript to emphasize Bonferroni-corrected results for analyses involving 21 tests or fewer. The BH-FDR values are still provided in the Supplementary Tables for transparency and comparability with prior studies. Importantly, incorporating Bonferroni correction did not alter our key conclusions: the principal associations highlighted in the main text remain significant, while nominal findings that did not meet Bonferroni significance have been moved to the Supplement.

Revised Text (Methods): "We corrected for multiple testing using the Bonferroni method, applying a significance threshold of $P < 0.00238$ ($0.05/21$ trait pairs). We also applied the Benjamini-Hochberg false-discovery rate (BH-FDR) procedure across all tests, considering $q < 0.05$ statistically significant, while associations with $P < 0.05$ but $q \geq 0.05$ are reported as suggestive."

Minor Comments:

- The figure text size is too small to be read.

Response: We thank the reviewer for highlighting this issue. We have increased the font sizes of all relevant figure elements, including axis labels, legends, annotations, and panel labels, to ensure readability in both print and digital formats. In addition, all figures are now provided as fully editable PowerPoint files to facilitate inspection and adjustment during production.

- Several labels on Figure 4 are misaligned.

Response: We thank the reviewer for noting the misaligned labels in Figure 4. All labels have now been corrected and uniformly aligned in the revised figure to ensure clarity and visual consistency.

• What do the colors bars in Figure 3i-k represent? The legend mentions the dashed line indicates nominal significance.

Response: We thank the reviewer for noting this ambiguity. We have now clarified the meaning of the bar colors in Figure 3i–k. In the revised legend, we explicitly state that the bars represent $-\log_{10}(P)$ values from MAGMA tissue enrichment; red bars indicate tissues with enrichment surpassing the nominal significance threshold, while blue bars indicate enrichment values that do not reach nominal significance. The dashed vertical line marks $P \leq 0.05$.

Reviewer #3 Remarks on code availability:

The provided github provides example scripts for the analyses run. For example, the LDSC contains the example of munging and running the cross-trait LDSC correlation between intelligence and T1D. The scripts are either provided as R/Rmd or bash scripts to be submitted to a SLURM scheduler. The code that I examined appeared correct. As the paper does not propose novel software I did not download or test the code.

Response: We thank the reviewer for carefully examining the provided scripts and for the positive assessment of our code availability and implementation. As noted, our GitHub repository includes reproducible examples for all major analyses (e.g., LDSC, MAGMA, SMR, conjFDR, SCAVENGE), with scripts provided in R, R Markdown, and Bash formats compatible with SLURM-based HPC environments. We appreciate the reviewer's confirmation that the code is correct and appropriately documented.

References

1. Lorincz-Comi, N. *et al.* (2024) MRBEE: A bias-corrected multivariable Mendelian randomization method. *HGG Adv* 5, 100290. 10.1016/j.xhgg.2024.100290
2. Giedd, J.N. *et al.* (2006) Puberty-related influences on brain development. *Molecular and cellular endocrinology* 254-255, 154–162. 10.1016/j.mce.2006.04.016
3. Luna, B. and Sweeney, J.A. (2001) Studies of brain and cognitive maturation through childhood and adolescence: a strategy for testing neurodevelopmental hypotheses. *Schizophr Bull* 27, 443–455. 10.1093/oxfordjournals.schbul.a006886
4. Goyal, M.S. and Raichle, M.E. (2018) Glucose Requirements of the Developing Human Brain. *J Pediatr Gastroenterol Nutr* 66 Suppl 3, S46–s49. 10.1097/mpg.0000000000001875
5. Vannucci, R.C. and Vannucci, S.J. (2000) Glucose metabolism in the developing brain. *Semin Perinatol* 24, 107–115. 10.1053/sp.2000.6361
6. Mauras, N. *et al.* (2021) Impact of Type 1 Diabetes in the Developing Brain in Children: A Longitudinal Study. *Diabetes Care* 44, 983–992. 10.2337/dc20-2125
7. Stanisławska-Kubiak, M. *et al.* (2024) Brain functional and structural changes in diabetic children. How can intellectual development be optimized in type 1 diabetes? *Ther Adv Chronic Dis* 15, 20406223241229855. 10.1177/20406223241229855
8. Cerolsaetti, K. *et al.* (2019) Genetics Coming of Age in Type 1 Diabetes. *Diabetes Care* 42, 189–191. 10.2337/dci18-0039
9. Bray, N.J. and O'Donovan, M.C. (2019) The genetics of neuropsychiatric disorders. *Brain Neurosci Adv* 2. 10.1177/2398212818799271

10. Lindbohm, J.V. *et al.* (2022) Immune system-wide Mendelian randomization and triangulation analyses support autoimmunity as a modifiable component in dementia-causing diseases. *Nat Aging* 2, 956–972. 10.1038/s43587-022-00293-x
11. Duan, J. *et al.* (2019) From Schizophrenia Genetics to Disease Biology: Harnessing New Concepts and Technologies. *Journal of Psychiatry and Brain Science* 4, e190014. 10.20900/jpbs.20190014
12. Zhu, B. *et al.* (2024) Single-cell transcriptomic and proteomic analysis of Parkinson's disease brains. *Science translational medicine* 16, eabo1997. 10.1126/scitranslmed.abo1997

POINT-BY-POINT RESPONSE TO REVIEWER COMMENTS

We thank the reviewers for their thorough evaluation and constructive feedback on our manuscript, “*Shared genetic and neuroimmune architecture links type 1 diabetes with neurocognitive traits.*” We have carefully considered all comments and substantially strengthened the manuscript through additional analyses, clarifications, and focused revisions.

To facilitate review, we are submitting both a clean version of the revised manuscript and a marked version indicating all changes made in response to the reviewers.

In particular, in response to Reviewer #1, we clarified the interpretation of polygenic overlap across methods and explicitly evaluated the potential influence of GWAS power on conjunctive false discovery rate (conjFDR) locus discovery using direct GWAS power metrics, as suggested by the reviewer. Using these direct GWAS power metrics, we re-evaluated whether the number of conjFDR-discovered pleiotropic loci was related to GWAS power. Corresponding supplementary tables and figures have been added to document these analyses.

In response to Reviewer #3, we corrected the methodological language in the stratified LDSC (S-LDSC) section to ensure that multiple-testing correction and correlation terminology are applied only where appropriate, and that S-LDSC results are consistently described in terms of heritability enrichment.

All reviewer comments are summarized below and addressed point by point.

Reviewer #1, Comment 1: Interpretation of MiXeR vs LDSC

Reviewer comment (summary): The manuscript states that MiXeR reveals shared variants with same and opposite effect directions not captured by LDSC. However, Figure 2 does not clearly demonstrate higher polygenic overlap than LDSC, and MiXeR itself does not encode effect direction. The relationship between LDSC genetic correlation and MiXeR overlap should be clarified.

Response: We appreciate this important clarification and agree with the reviewer’s interpretation. MiXeR does not explicitly encode effect direction, and antagonistic pleiotropy is therefore not directly visualized by MiXeR alone. Rather, heterogeneous or opposing effects are inferred from the *discordance* between substantial MiXeR-estimated polygenic overlap and weak or inverse LDSC genetic correlations.

To avoid overinterpretation, we have revised the text to explicitly distinguish the complementary roles of the two methods. We now clarify that MiXeR quantifies the *extent* of shared causal variation irrespective of direction, whereas LDSC summarizes the *average directional alignment* of effects across the genome.

No new analyses were required; instead, the interpretation has been tightened and aligned more precisely with what is shown in Figure 2.

Manuscript changes (Results, MiXeR section):

Original text: “Importantly, the MiXeR-estimated overlaps demonstrate that substantial numbers of shared causal variants influence both T1D and neurocognitive traits, including variants with opposite effect directions, relationships that are not captured by genome-wide genetic correlation alone.”

Revised text: “MiXeR analyses revealed extensive sharing of causal variants between T1D and neurocognitive traits irrespective of effect direction. In several cases, substantial polygenic overlap co-occurred with weak or inverse genome-wide genetic correlations, indicating that shared variants may exert heterogeneous or opposing effects that are averaged out in LDSC-based correlation estimates.”

Reviewer #1, Comment 2: Relationship between conjFDR loci and GWAS power

Reviewer comment (summary): Supplementary Table 1 only reports sample size, which is not an adequate proxy for GWAS power. The claim that pleiotropic locus counts are unrelated to power is insufficiently supported. The reviewer requests inclusion of more direct power metrics (e.g., mean χ^2 , LDSC heritability Z-score) and re-evaluation of the relationship.

Response: We agree with the reviewer that GWAS sample size alone is an imperfect proxy for statistical power. In response, we have added a new **Supplementary Table 4** reporting two direct and commonly used GWAS power metrics: mean χ^2 statistics and LDSC heritability Z-scores. Using these metrics, we re-evaluated whether the number of conjFDR-discovered pleiotropic loci was related to GWAS power. We additionally included a new supplementary scatter plot (**Supplementary Fig. 2**) visualizing the relationship between pleiotropic locus counts and GWAS power, with Spearman rank correlation statistics reported.

These analyses demonstrate no strong or monotonic relationship between GWAS power—assessed using mean χ^2 statistics or LDSC heritability Z-scores—and the number of conjFDR loci identified. This supports our interpretation that pleiotropic locus counts primarily reflect trait-specific shared genetic architecture rather than differences *in statistical power*. *The Results text has been revised accordingly to avoid overstatement.*

Manuscript changes

- **Supplementary Table 4 (added):** GWAS power metrics and conjFDR pleiotropic locus counts for T1D–neurocognitive trait pairs, including mean χ^2 statistics and LDSC heritability Z-scores.
- **Supplementary Figure 2 (added):** Scatter plot showing the relationship between GWAS power, measured by LDSC heritability Z-score, and the number of loci jointly associated with type 1 diabetes (T1D) and each neurocognitive or neuroimmune trait at conjFDR < 0.05. Each point represents one external trait.
- **Results section (conjFDR paragraph):**

Original text (replaced): “Importantly, the number of conjFDR pleiotropic loci did not correspond to GWAS sample size or effective power (Supplementary Table 1).”

Revised text (new): “Importantly, the number of pleiotropic loci identified by conjFDR did not show a significant relationship with GWAS power, as assessed using LDSC heritability Z-scores (Supplementary Fig. 2; Suppl Table 4). Traits spanning a wide range of effective GWAS power exhibited both high and low degrees of pleiotropic overlap with T1D, indicating that conjFDR locus counts primarily reflect trait-specific shared genetic architecture rather than statistical power.”

Suppl Table 4: GWAS power metrics and conjFDR pleiotropic locus counts for T1D–neurocognitive trait pairs

Trait	N_conjFDR_loci	h ² _obs	h ² _obs_se	h ² _z
Multiple sclerosis	109	0.3286	0.0611	2.06
Myasthenia gravis	69	0.0426	0.0065	4.65
Intelligence	63	0.0428	0.0065	-3.95

Educational attainment	45	0.0424	0.0064	-2.33
Alzheimer's disease	29	0.0428	0.0064	1.52
Executive function	21	0.0420	0.0068	-3.21
Neuroticism	19	0.0428	0.0065	1.19
Bipolar disorder	13	0.0424	0.0064	-4.00
Migraine	9	0.0451	0.0063	2.54
Parkinson's disease	9	0.0428	0.0063	0.34
ALS	6	0.0429	0.0068	0.69
Autism spectrum disorder	6	0.0438	0.0070	-2.21
ADHD	1	0.0443	0.0064	-0.50
Insomnia	1	0.0422	0.0067	2.11

Supplementary Figure 2: GWAS power versus pleiotropic locus discovery by conjFDR. Scatter plot showing the relationship between GWAS power, measured by LDSC heritability Z-score, and the number of loci jointly associated with type 1 diabetes (T1D) and each neurocognitive or neuroimmune trait at conjFDR < 0.05. Each point represents one external trait.

Reviewer #1, Code Availability

Reviewer comment: The code is open-source and well-documented.

Response: We thank the reviewer for this assessment. No changes were required.

Reviewer #2

Reviewer comment (summary): Referencing should be double-checked; an example citation mismatch was noted.

Response: We thank the reviewer for pointing this out. We have systematically reviewed all in-text citations and reference assignments throughout the manuscript and corrected several misaligned references, including the example noted by the reviewer where a citation referring to human neocortical tissue was corrected to reference appropriate peripheral immune cell datasets. All references now accurately reflect the statements they support.

Reviewer #3

Reviewer comment (summary): Language regarding multiple testing correction appears in both the S-LDSC and cross-trait LDSC sections, although S-LDSC analyses were not adjusted for multiple testing.

Response: We thank the reviewer for identifying this inconsistency and agree with the concern. The multiple-testing correction language has now been removed from the S-LDSC section and retained only in the cross-trait LDSC section, where it is appropriate. The S-LDSC text has been revised to refer specifically to heritability enrichment rather than correlation or multiple-testing-adjusted significance.

Revised text (methods): "S-LDSC analyses were performed using S-LDSC v1.0.1 with the baseline-LD model v2.2 and European reference LD scores from the 1000 Genomes Project Phase 3. Heritability enrichment was quantified as the proportion of SNP-heritability explained by each annotation divided by the proportion of SNPs overlapping that annotation. Statistical uncertainty was assessed using standard errors provided by the S-LDSC framework, and nominal p-values are reported to summarize evidence for enrichment. Results are interpreted as patterns of heritability enrichment across annotations."

POINT-BY-POINT RESPONSE TO REVIEWER COMMENTS

We thank the reviewer for their careful evaluation and constructive feedback on our manuscript, “*Shared genetic and neuroimmune architecture links type 1 diabetes with neurocognitive traits.*” We have carefully considered the reviewer’s concerns and implemented targeted clarifications and corrections to ensure complete accuracy and transparency of the supplementary materials and GWAS power analyses.

To facilitate review, we are submitting both a **clean revised manuscript** and a **marked version** indicating all changes made in response to the reviewers. Reviewer comments are summarized below and addressed point by point.

Response to Reviewer #1, Comment: GWAS power metrics and SNP-heritability Z-statistics

Reviewer concern (paraphrased). The reviewer notes two related issues in the revised submission:

1. **Location and completeness of GWAS power metrics.** The manuscript text indicates that additional GWAS power metrics—specifically **mean χ^2 statistics**—were added to the supplementary materials. However, these values were not present in the referenced supplementary tables. This raises concern that some described revisions may not have been fully implemented or clearly presented.
2. **Correctness of SNP-heritability Z-statistics.** The reviewer further observes that several reported **heritability Z-statistics** appear inconsistent with the ratio of estimate to standard error, and that approximately half of the reported Z-values are negative, which would be unexpected for well-powered traits. The reviewer therefore requests verification and correction of these statistics to ensure consistency with **LDSC conventions**.

Response to Reviewer #1. We thank the reviewer for highlighting important issues regarding (i) the **location and presentation of GWAS power metrics** and (ii) the **correctness of SNP-heritability Z-statistics** reported in the supplementary materials.

1. Location and presentation of GWAS power metrics. We clarify that **Supplementary Tables 1 and 2** were intentionally left unaltered, as they serve distinct and predefined purposes within the manuscript:

- **Supplementary Table 1** summarizes GWAS datasets (source, ancestry, and sample size).
- **Supplementary Table 2** reports genome-wide genetic correlation statistics.

Because neither table is designed to evaluate GWAS statistical power, the reviewer-requested metrics—mean χ^2 statistics and LDSC SNP-heritability Z-scores—have been compiled in a newly created **Supplementary Table 4**, which is explicitly dedicated to assessing the relationship between GWAS power and conjFDR pleiotropic locus discovery across T1D–neurocognitive trait pairs.

All manuscript text and table citations have been revised to ensure that references to GWAS power metrics now correctly point to Supplementary **Table 4**, eliminating ambiguity regarding their location.

This clarification concerns presentation only and does not affect analytical results or conclusions.

2. Verification and correction of SNP-heritability Z-statistics. The reviewer noted that several previously reported Z-statistics appeared inconsistent with the ratio of estimate to standard error and that multiple values were negative, which would be unexpected for well-powered traits.

Upon re-examination, we identified a **column-labeling error** in the prior submission in which the **genetic-correlation Z-statistic (rg Z)** had inadvertently been presented in the column labeled as the **heritability Z-statistic**. Because rg Z-scores may be positive or negative depending on the direction of genetic correlation, this mislabeling created the appearance of implausible negative “heritability” Z-values.

This issue has now been fully corrected.

In the revised **Supplementary Table 4**, SNP-heritability Z-statistics are computed strictly according to LDSC conventions:

$$Z_{h^2} = \frac{h^2}{SE(h^2)}.$$

All values were **recomputed directly from LDSC output logs** and verified for internal consistency.

Following correction:

- Heritability Z-statistics are fully consistent with **estimate/SE**.
- Previously negative values are now confirmed to have originated from **rg Z-scores**, not heritability Z-scores.
- GWAS power metrics are now **accurately and transparently reported**.

3. Relationship between GWAS power and conjFDR locus discovery. Using the corrected metrics in **Supplementary Table 4**, we evaluated whether **GWAS power explains variation** in the number of conjFDR-identified pleiotropic loci.

Across the analyzed traits:

- **Mean χ^2** showed only a **nominal positive correlation** with conjFDR locus counts in this limited trait set.
- In contrast, **LDSC heritability Z-scores**, which reflect confounding-adjusted polygenic signal and are widely used as a more direct LDSC-based measure of effective GWAS power, showed **no association** with conjFDR locus counts.

Importantly, mean χ^2 and heritability Z-scores capture distinct properties of GWAS signal: mean χ^2 reflects overall test-statistic inflation influenced by sample size, polygenicity, and residual confounding, whereas heritability Z-scores quantify true detectable SNP-heritability after intercept-based correction. The lack of concordance across independent GWAS power metrics, together with the small number of traits analyzed and sensitivity to individual outliers, indicates that pleiotropic locus discovery is not primarily determined by GWAS statistical power. Rather, the number of conjFDR loci more plausibly reflects trait-specific shared genetic architecture with T1D.

4. Impact on conclusions. These revisions relate solely to:

- clarification of **supplementary table structure**,
- correction and verification of **heritability Z-statistics**, and
- transparent reporting of **GWAS power analyses**.

They **do not alter**:

- GWAS power comparisons,

- the absence of a **robust relationship** between GWAS power and conjFDR locus counts, or
- the biological interpretation of **shared genetic architecture underlying T1D–neurocognitive overlap**.

All analytical results and scientific conclusions therefore remain **unchanged**.

Suppl Table 4: GWAS power metrics and conjFDR pleiotropic locus counts for T1D (520 580 (18 942 cases)—neurocognitive trait pairs.

Trait	n_conjFDR_loci	Mean χ^2	h^2_{obs}	$h^2_{obs_se}$	h^2_Z
Bipolar disorder	13	1.5887	0.0716	0.0028	25.57
Multiple sclerosis	109	2.1596	1.5271	0.2520	6.06
Autism spectrum disorder	6	1.1998	0.2521	0.0217	11.62
Myasthenia gravis	69	1.0988	0.0095	0.0016	5.94
Educational attainment	45	1.6447	0.0918	0.0033	27.82
Executive function	21	1.8415	0.0931	0.0042	22.17
Insomnia	1	1.3659	0.0456	0.0021	21.71
Migraine	9	1.2492	0.0220	0.0015	14.67
Intelligence	63	2.0450	0.1883	0.0069	27.29
Alzheimer's disease	29	1.2735	0.0189	0.0031	6.10
Neuroticism	19	1.8629	0.1145	0.0041	27.93
Parkinson's disease	9	1.0893	0.0059	0.0006	9.83
ADHD	1	1.4490	0.0948	0.0045	21.07
ALS	6	1.1313	0.0382	0.0044	8.68

ALS, Amyotrophic lateral sclerosis; ADHD, Attention deficit/hyperactive disorder; T1D, Type 1 diabetes

Supplementary Figure 2 | GWAS power versus pleiotropic locus discovery by conjFDR.

Scatter plots showing the relationship between GWAS power and the number of loci jointly associated with type 1 diabetes (T1D) and each external neurocognitive or neuroimmune trait at conjFDR < 0.05. **(a)** Mean χ^2 statistic from LDSC for the external trait GWAS versus conjFDR pleiotropic locus count. **(b)** LDSC SNP-heritability Z-score for the external trait GWAS, computed as $Z(h^2) = h^2_{obs} / SE(h^2_{obs})$, versus conjFDR pleiotropic locus count. Each point represents one external trait; Pearson correlation coefficients (r) and two-sided P values are shown.